# CURE: Context-driven Diffusion with Progressive Expansion for Single Domain Generalization in Time Series Classification

**Yuhang Pei** [* 1] **Fanchun Meng** [* 1] **Wenrui Wu** [2] **Tao Ren** [1] **Yifan Wang** [3] **Wei Ju** [2] **Chao Zheng** [4] **Xiao Luo** [5]

## Abstract

This paper studies the problem of single domain generalization in time series classification, which aims to learn a generalized time series classification model using a single source domain. This problem is highly challenging due to unreliable supervision from domain scarcity. Although current approaches employ generative models for data augmentation, these synthesized samples often suffer from low diversity and intrinsic noise, leading to weak generalization ability. Towards this end, we propose a novel approach named Context-driven Diffusion with Progressive Expansion (CURE) for single domain generalization in time series classification. The core of our CURE is to generate semantic-aware and semantic-free contexts that jointly guide a conditional diffusion model for informative data expansion. In particular, our CURE first conducts representation disentanglement to extract semantic-aware and semantic-free representations from source data. To enhance generalizability through data synthesis, we not only retrieve reference time series trajectories with similar semantics for semantic-aware contexts, but also utilize adversarial strategies to learn semantic-free contexts. These contexts are integrated as joint conditions for a diffusion model, enabling diverse and reliable virtual data. To enhance expansion adaptability and stable optimization, we progressively update our semantic-free contexts via a memory bank and measure boundary properties for dynamic data filtering. Comprehensive experiments on benchmark datasets validate the effectiveness of CURE.

---

[*]Equal contribution [1]Software College, Northeastern University, Shenyang, China [2]Peking University, Beijing, China [3]University of International Business and Economics, Beijing, China [4]University of Southampton, United Kingdom [5]Department of Statistics, University of Wisconsin–Madison, USA. Correspondence to: Yifan Wang <yifanwang@uibe.edu.cn>.

*Proceedings of the 43$^{rd}$ International Conference on Machine Learning*, Seoul, South Korea. PMLR 306, 2026. Copyright 2026 by the author(s).

## 1. Introduction

Time series Classification (TSC) serves as a pivotal task in various fields, ranging from finance analytics (Zheng et al., 2021), healthcare monitoring (Schirrmeister et al., 2017; Rajkomar et al., 2018) and human activity recognition (Wang et al., 2019; Chen et al., 2021). The goal of TSC is to assign a class label to an observed time-series data and assumes that the data follows an independent and identically distributed (i.i.d.) distribution. Nevertheless, this assumption is typically violated in real-world time series data, where the observations are temporally dependent and the underlying data distribution could vary because of factors such as data acquisition from different sources or changes in environmental conditions. This distribution shift between the training (source) and test (target) domains can lead to significant deterioration in model performance (Zhou et al., 2022).

Recently, there has been growing interest in Domain Generalization (DG) for TSC tasks (Wu et al., 2025; Lu et al., 2024). Data-driven approaches aim to manipulate the input data to enhance the diversity of the data distribution. These methods include techniques like data augmentation (Zhang et al., 2018; Volpi et al., 2018) and distribution intervention (Wu et al., 2022b; Liu et al., 2022). On the other hand, learning-based approaches aim to enhance generalization by adjusting the model's training process. This includes techniques such as (causal) invariant learning (Creager et al., 2021; Li et al., 2022), which enforces consistent causal relationships across domains, incorporating model regularization (Zhao et al., 2020) to prevent overfitting and improve robustness, or modifying the model architecture (Zhu et al., 2021; Yang et al., 2023a) to better capture complex patterns and improve adaptability to diverse data distributions.

Despite the effectiveness of these methods, most of them rely on multiple source domains to generalize effectively to unseen target domains. In the TSC tasks, as environmental dynamics evolve and the diversity of the potential test domains expands (Deng et al., 2024; Zhang et al., 2025), the limited variability within the available training data may no longer be sufficient to generalize effectively to the broad range of unseen environments. This challenge has led to the growing interest in Single Domain Generalization (SDG) (Qiao et al., 2020; Wang et al., 2021; Xu et al., 2025)

for TSC tasks, where the focus shifts to enabling models to generalize effectively when trained only on a single domain.

In fact, we argue that achieving SDG in the TSC task remains challenging due to the following two key issues: ❶ *Limited explicit semantic guidance during the augmentation process.* Most of these methods fail to disentangle causal invariant category information from domain-specific features for the time series data. As a result, clear semantic supervision is limited during the augmentation process, and the augmented data fails to maintain the necessary semantic structure, which hinders the model's ability to generalize effectively across domains. ❷ *Diversity collapse due to inferior augmentation data.* During the augmentation process, intrinsic noise may arise from the inadequacy of the feature space. Simply selecting boundary samples as anchors for augmentation may result in a narrow focus on marginal instances, which can ultimately lead to a problematic collapse in the diversity of the augmented data.

Towards this end, in this paper, we propose a novel **C**ontext-driven Diff**u**sion with P**r**ogressive **E**xpansion (**CURE**) for single domain-generalized time series classification, which formulates the conditional diffusion model with the disentangled prompts as explicit semantic guidance to generate out-of-distribution time series data. Specifically, to capture distinct distributional characteristics over time while maintaining invariant class features, we decompose the learned features of each time-series instance into semantic-aware and semantic-free information. Then, we incorporate these features as contexts to provide essential prompts for the conditional diffusion model. On the one hand, we encourage category semantic consistency between the generated time-series data and their corresponding category prompts. On the other hand, we minimize style similarity between newly generated domains and the existing semantic-free domain contexts stored in the memory bank. Furthermore, by updating the domain prompt memory bank and filtering out the boundary samples, we can progressively expand the diversity of training time series data in a self-paced learning paradigm for the single domain-generalized TSC task.

In a nutshell, the contribution of our paper can be summarized as follows:

❶ *New Perspective:* We highlight the limited variability within the time series data and propose a novel data-centric perspective to effectively tackle single domain-generalized time series classification.

❷ *Novel Methodology:* We propose CURE, which leverages the disentangled category and domain context as explicit guidance to adversarially balance the trade-off between diverse domain styles and key categorical features in generated time series data.

❸ *Extensive Experiments:* We conduct extensive experiments to evaluate the performance of CURE. The result demonstrates that our CURE achieves superior performance over baselines.

## 2. Preliminary

### 2.1. Problem Definition

Let a time series training dataset be denoted as $\mathcal{D}^{tr} = \{(\boldsymbol{x}_i, y_i)_{i=1}^N\}$ consisting of $N$ time series data, where $\boldsymbol{x}_i \in \mathcal{X} = \mathbb{R}^{D \times L}$ represents $L$-length time series data with $D$ dimensional features at each time step, and $y_i \in \mathcal{Y} = \{1, \ldots, C\}$ is the corresponding label across $C$ classes. We denote the joint distribution of the training dataset as $P_{tr}(\boldsymbol{x}, y|e)$, in which $e \in \mathcal{E}$ denotes the environment within the entire set $\mathcal{E}$. The distribution shift induces a discrepancy between the training and test distributions, i.e., $P_{tr}(\boldsymbol{x}_i, y_i) \neq P_{te}(\boldsymbol{x}_i, y_i)$, while the feature and label space remain consistent across domains, i.e., $\mathcal{X}_{tr} = \mathcal{X}_{te}$ and $\mathcal{Y}_{tr} = \mathcal{Y}_{te}$. In the single domain generalization setting, we assume the training data is collected from a single source domain, i.e., $|\mathcal{E}| = 1$. The objective is to learn a generalized TSC model which can handle distribution shifts.

### 2.2. Denoising Diffusion Probabilistic Models

Diffusion probabilistic models (Sohl-Dickstein et al., 2015; Yang et al., 2023c) formulate the generative process as a latent variable model $p_\theta(\boldsymbol{x}^0) := \int p_\theta(\boldsymbol{x}^{0:T}) d\boldsymbol{x}^{1:T}$, in which $\boldsymbol{x}^1, \ldots, \boldsymbol{x}^T$ represent latent variables over $T$ steps. As a widely known diffusion model, DDPM (Ho et al., 2020; Croitoru et al., 2023) is extensively applied in probabilistic generation tasks, which is composed of a forward diffusion process and a reverse denoising process. Formally, the forward process gradually adds Gaussian noise to $\boldsymbol{x}^0$ following a Markov chain over $T$ steps, described as:

$$q(\boldsymbol{x}^{1:T}|\boldsymbol{x}^0) := \prod_{t=1}^T q(\boldsymbol{x}^t|\boldsymbol{x}^{t-1}),$$
$$q(\boldsymbol{x}^t|\boldsymbol{x}^{t-1}) := \mathcal{N}(\sqrt{1-\beta_t}\boldsymbol{x}^{t-1}, \beta_t \boldsymbol{I}), \quad (1)$$

where $\beta_t \in (0, 1)$ denotes the variance of Gaussian noise included at the $t$-th step. The endpoint of the diffusion process follows the Gaussian prior, i.e., $\boldsymbol{x}^t \sim \mathcal{N}(0, \boldsymbol{I})$. In practice, $\boldsymbol{x}^t$ at step $t$ can be directly derived from $\boldsymbol{x}^0$ with a single step:

$$q(\boldsymbol{x}^t|\boldsymbol{x}^0) = \mathcal{N}(\sqrt{\bar{\alpha}_t}\boldsymbol{x}^0, (1-\bar{\alpha}_t)\boldsymbol{I}),$$
$$\boldsymbol{x}^t = \sqrt{\bar{\alpha}_t}\boldsymbol{x}^0 + (1-\bar{\alpha}_t)\boldsymbol{\epsilon}, \quad (2)$$

where $\alpha_t := 1 - \beta_t$ and $\bar{\alpha}_t := \prod_{i=1}^t \alpha_i$. And $\boldsymbol{\epsilon} \sim \mathcal{N}(0, \boldsymbol{I})$ here is the sampled noise. The reverse denoising process parameterized by $p_\theta(\boldsymbol{x}^{t-1}|\boldsymbol{x}^t)$ aims to iteratively remove the injected noise:

$$p_\theta(\boldsymbol{x}^{t-1}|\boldsymbol{x}^t) := \mathcal{N}(\boldsymbol{\mu}_\theta(\boldsymbol{x}^t, t), \Sigma_\theta(\boldsymbol{x}^t, t)), \quad (3)$$

where $\boldsymbol{\mu}_\theta(\boldsymbol{x}^t, t)$ is parameterized by a neural network with parameters $\theta$ and $\Sigma_\theta(\boldsymbol{x}^t, t)$ is typically fixed, formulated as:

$$\boldsymbol{\mu}_\theta(\boldsymbol{x}^t, t) := \frac{1}{\sqrt{\alpha_t}} \left( \boldsymbol{x}^t - \frac{\beta_t}{\sqrt{1 - \bar{\alpha}_t}} \boldsymbol{\epsilon}_\theta \right),$$
$$\Sigma_\theta(\boldsymbol{x}^t, t) := \frac{1 - \bar{\alpha}_{t-1}}{1 - \bar{\alpha}_t} \beta_t \boldsymbol{I}. \tag{4}$$

The diffusion model is usually trained by maximizing the evidence lower bound (ELBO) of the log-likelihood, simplified as:

$$\mathcal{L}_\epsilon = \mathbb{E}_{\boldsymbol{x}^0 \sim q(\boldsymbol{x}^0), \boldsymbol{\epsilon} \sim \mathcal{N}(0, \boldsymbol{I}), t} ||\boldsymbol{\epsilon} - \boldsymbol{\epsilon}_\theta(\boldsymbol{x}^t, t)||^2. \tag{5}$$

Note that to synthesize diverse time series under specific guidance, we consider a conditional distribution $p_\theta(\boldsymbol{x}^0|\boldsymbol{p})$ with the condition $\boldsymbol{p}$. The conditional diffusion model can be defined by $q(\boldsymbol{x}^t|\boldsymbol{x}^{t-1}, \boldsymbol{p})$ and $p_\theta(\boldsymbol{x}^{t-1}|\boldsymbol{x}^t, \boldsymbol{p})$ for the forward and reverse process. The learned injected noise can be further represented as $\boldsymbol{\epsilon}_\theta(\boldsymbol{x}^t, t, \boldsymbol{p})$.

# 3. The Proposed CURE

## 3.1. Framework Overview

This paper focuses on the challenge of limited diversity in single domain-generalized TSC task and proposes a novel progressive diversity expansion framework CURE. The fundamental principle of our framework is to introduce explicit guidance into the diffusion model so as to promote diverse distribution generation while maintaining semantic consistency. In particular, given the input time-series instances from a single domain, we learn to decompose the invariant semantics-aware information and the semantic-free domain contexts. Then we leverage them as prompts to construct the explicit guidance for the diffusion model. Furthermore, we learn the new domain prompt to update the prompt memory bank and relax to filter out boundary samples to progressively expand the distribution diversity. The overview of our CURE is illustrated in Figure 1 and we present each component as follows.

## 3.2. Representation Disentanglement for Source Data Exploration

To provide precise guidance for diffusion-based generation, we disentangle each time-series instance into an invariant category feature that preserves semantic information and a specific context that encodes domain contextual variations.

**Semantic-aware Context Exploration.** Effective handling of distributional shifts requires capturing category invariances that are consistent across domains. Specifically, given the time instance $\boldsymbol{x}_i$, we employ an encoder $f_\psi(\cdot)$ with parameter $\psi$ to extract the domain-invariant feature, i.e., $\boldsymbol{c}_i = f_\psi(\boldsymbol{x}_i)$. Then, we employ a supervised contrastive

learning (Khosla et al., 2020) to enforce the feature to retain class semantics. The objective can be defined as:

$$\mathcal{L}_{sup} = -\sum_{i=1}^{N} \frac{1}{|\mathcal{P}(i)|} \sum_{l \in \mathcal{P}(i)} \log \frac{\exp(\boldsymbol{c}_i \star \boldsymbol{c}_l / \tau)}{\sum_{j \in \mathcal{A}(i)} \exp(\boldsymbol{c}_i \star \boldsymbol{c}_j / \tau)}, \tag{6}$$

where $\mathcal{A}(i) = \{1, \ldots, N\}/\{i\}$ includes all indices except $i$ and we set the positive set $\mathcal{P}(i) = \{l | l \in \mathcal{A}(i), y_l = y_i\}$ as indices with same class label. Here $\star$ denotes cosine similarity between two vectors and $\tau$ is the temperature parameter.

**Semantic-free Context Preservation.** To preserve complementary information related to the semantic-free temporal context, we introduce another encoder $f_\phi$ with parameter $\phi$ to extract the domain-specific feature, i.e., $\boldsymbol{s}_i = f_\phi(\boldsymbol{x}_i)$. Building upon these representations, we impose instance-wise contrastive learning to capture rich intra-instance distributional variations. Furthermore, we incorporate a class invariance constraint with a gradient reversal layer (GRL) (Ganin & Lempitsky, 2015; Zhang et al., 2025) $R(\cdot)$ to suppress class-related signals:

$$\mathcal{L}_{sp} = \sum_{i=1}^{N} \left[ \ell_{ce}(y_i, h(R(\boldsymbol{s}_i))) - \alpha \frac{\exp(\boldsymbol{s}_i \star \boldsymbol{s}_i^+ / \tau)}{\sum_{j \neq i} \exp(\boldsymbol{s}_i \star \boldsymbol{s}_j / \tau)} \right], \tag{7}$$

where $\boldsymbol{s}_i^+$ denotes extracted features from the augmentation with the time-step masking (Yue et al., 2022). $h(\cdot)$ is the class discriminator and $\alpha$ is the trade-off parameter between two losses. To further prevent redundancy, we characterize the dependence between two extracted representations with a distance correlation constraint to maintain distinct separation between these two subspaces, defined as:

$$\mathcal{L}_{ind} = dCor(\boldsymbol{C}, \boldsymbol{S}) = \frac{dCov(\boldsymbol{C}, \boldsymbol{S})}{\sqrt{dVar(\boldsymbol{C}) \cdot dVar(\boldsymbol{S})}}, \tag{8}$$

where $\boldsymbol{C} = [\boldsymbol{c}_1, \ldots, \boldsymbol{c}_N]$ and $\boldsymbol{S} = [\boldsymbol{s}_1, \ldots, \boldsymbol{s}_N]$ denote the packed invariant semantic-aware and semantic-free context feature matrix. $dVar(\cdot)$ measures the distance variance of each matrix and $dCov(\cdot)$ calculates the distance covariance between two matrices.

## 3.3. Context-driven Diffusion Model for High-quality Data Expansion

Given the disentangled invariant semantic-aware information and semantic-free domain context, we employ them as conditions (Narasimhan et al., 2024; Zhou et al., 2025; Koulischer et al., 2025; Zhang et al., 2025) for the diffusion model to generate wide variety of diverse time series data.

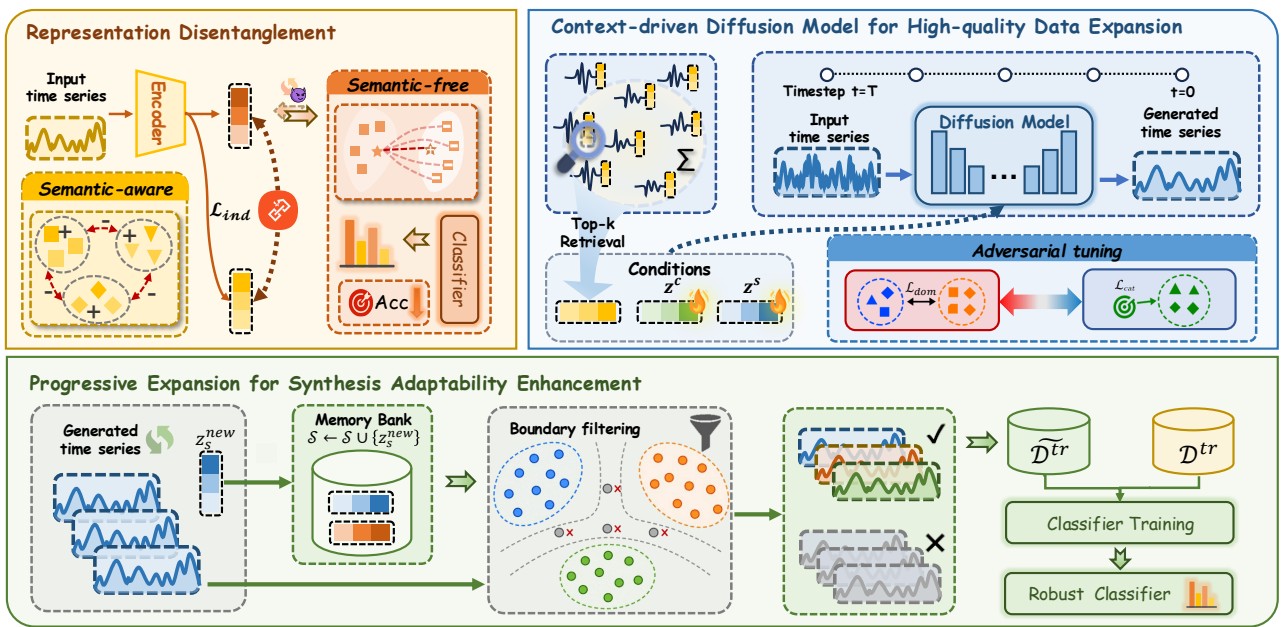

*Figure 1.* Overview of the Progressive Diversity Expansion Framework (CURE). The framework comprises three modules: (1) Representation Disentanglement extracts semantic-aware and semantic-free contexts. (2) Context-driven Diffusion Model uses these features to generate diverse time series through adversarial tuning. (3) Progressive Diversity Expansion iteratively updates a Memory Bank and filters boundary samples to enrich the augmented dataset for robust classification.

**Conditional Contexts Construction.** Since similar time series behaviors inherently convey rich semantic information, we measure the similarity of category features across time series instances and retrieve the most similar examples. These retrieved instances are then used to construct a semantic-aware condition to guide the synthesis of new time series data. Specifically, for each $\boldsymbol{x}_i$, the reference $k$ time series trajectories can be retrieved as:

$$r_k(\boldsymbol{x}_i) := \inf\{r : |\mathcal{B}(\boldsymbol{x}_i, r) \cap \mathcal{X}| \geq k\},$$
$$\mathcal{B}(\boldsymbol{x}_i, r) := \{\boldsymbol{x}_j \in \mathcal{X} : \boldsymbol{c}_i \star \boldsymbol{c}_j \leq r\}, \quad (9)$$

where $r_k(\boldsymbol{x}_i)$ denote the radius of the retrieved $k$ nearest trajectories. Therefore, the retrieved set can be defined as:

$$\mathcal{N}_k(\boldsymbol{x}_i) := \mathcal{B}(\boldsymbol{x}_i, r_k(\boldsymbol{x}_i)) \cap \mathcal{X}. \quad (10)$$

These retrieved time series instances are then aggregated to summarize the invariant temporal patterns of the category:

$$\boldsymbol{c}_{\mathcal{N}_k(\boldsymbol{x}_i)} = \frac{1}{|\mathcal{N}_k(\boldsymbol{x}_i)|} \sum_{j \in \mathcal{N}_k(\boldsymbol{x}_i)} \boldsymbol{c}_j. \quad (11)$$

We further use invariant category information and semantic-free domain context to construct a set of guidance as:

$$\boldsymbol{p} = [\boldsymbol{c}_{\mathcal{N}_k(\boldsymbol{x}_i)}, *, \dagger], * \in [1, C], \dagger \in [|\mathcal{S}| + 1, |\mathcal{S}| + M], \quad (12)$$

where $*$ and $\dagger$ are the placeholders for the corresponding class and domain. We employ a learnable category

prompt $\boldsymbol{z}_c$ and domain prompt $\boldsymbol{z}_s \in \mathcal{S}$ as the condition, i.e., $\boldsymbol{p} = [\boldsymbol{c}_{\mathcal{N}_k(\boldsymbol{x}_i)}, \boldsymbol{z}_c, \boldsymbol{z}_s]$. Note that here we jointly leverage $C$ category prompts and other $M$ domain prompts out of memory bank $\mathcal{S}$ to construct the condition sets.

**Adversarial Tuning for Diffusion Model.** Given these domain conditions, we leverage them as explicit guidance and to adversarially generate a sufficiently diverse range of time series data across different domains (Xu et al., 2025; Cui et al., 2025). On the one hand, we aim to maximize the consistency between the generated sample $\boldsymbol{x}_{i'}^0$ and the category semantics $\boldsymbol{c}$, which can be defined as:

$$\mathcal{L}_{cat} = -\frac{1}{|\mathcal{P}(i')|} \sum_{l \in \mathcal{P}(i')}$$
$$\log \frac{\exp(f_\psi(\boldsymbol{x}_{i'}^0) \star \boldsymbol{c}_l / \tau)}{\sum_{j \in \mathcal{A}(i')} \exp(f_\psi(\boldsymbol{x}_{i'}^0) \star \boldsymbol{c}_j / \tau)}, \quad (13)$$

On the other hand, we aim to minimize the total style similarity between $\boldsymbol{x}_{i'}^0$ and the previous domain context $\boldsymbol{s}$:

$$\mathcal{L}_{dom} = -\sum_{s=1}^{|\mathcal{S}|} -\log \frac{\exp(f_\phi(\boldsymbol{x}_{i'}^0) \star \boldsymbol{z}_s / \tau)}{\sum_{s'=1}^{|\mathcal{S}|} \exp(f_\phi(\boldsymbol{x}_{i'}^0) \star \boldsymbol{z}_{s'} / \tau)}. \quad (14)$$

The final adversarial tuning loss can be:

$$\mathcal{L}_{adv}(\boldsymbol{x}_{i'}^0) = \mathcal{L}_{cat} + \beta \mathcal{L}_{dom}, \quad (15)$$

where $\beta$ is a hyper-parameter. Note that to guide the diffusion process for the adversarial tuning, we first obtain the

*predicted* time series instance $\hat{x}_{i'}^0$ from the step $t$ with low noise and the loss gradient is only back-propagated from the $t$-th to the 0-th step:

$$\hat{x}_{i'}^0 = \frac{1}{\sqrt{\bar{\alpha}_t}}(\hat{x}_{i'}^t - \sqrt{1 - \bar{\alpha}_t}\epsilon_\theta(\hat{x}_{i'}^t, t, p)). \quad (16)$$

### 3.4. Progressive Expansion for Synthesis Adaptability Enhancement

To enrich the synthesized data beyond the constraints of a single-source domain, we propose a progressive distribution diversity expansion scheme, which updates the domain prompt memory bank and filters out the boundary samples.

**Dynamic Contexts with Memory Bank Updating.** A primary challenge in single-domain scenarios is that the lack of cross-domain variability often confines the diffusion model to generating samples with styles overly similar to the source data. To foster distributional diversity (Zhang et al., 2025; Liu et al., 2024), we establish a dynamic memory bank (Koulischer et al., 2025; Liu et al., 2025; Xu et al., 2025) that serves as a repository for virtual environment prototypes. Initially, for the extracted $\{s_i\}_{i=1}^N$, we compute their average to initialize the domain prompts of the source domain. After a certain number of prompt tuning iterations, the memory bank $|\mathcal{S}|$ is updated with newly learned domain prompts, which progressively expand the coverage of domain variations, enabling the diffusion model to explore a broader distribution space while maintaining semantic consistency:

$$\mathcal{S} \leftarrow \mathcal{S} \cup \{z_s\}. \quad (17)$$

By extending the memory bank with $M$ such learnable prompts, the final capacity becomes $|\mathcal{S}| = M+1$, providing a rich set of conditions for diverse data expansion.

**Dynamic Filtering with Boundary Exploration.** Although the generated time series data offers more domain distribution diversity, there still exist some unreliable instances that constitute a detriment to the latter training, especially when they lie near the decision boundaries where distribution discrimination is ambiguous. Inspired by the silhouette score of clustering (Yu et al., 2023), we calculate the intra-domain distance $d_i^I$ and inter-domain $d_i^N$ as:

$$d_i^I = \|s_i - z_{\zeta(s_i)}\|_2, \quad d_i^N = \|s_i - z_{\zeta'(s_i)}\|_2, \quad (18)$$

where $\zeta(\cdot)$ denotes the corresponding domain for the time series instance, and $\zeta'(s_i) = \arg\min_{s \in \{1,...,|\mathcal{S}|\} \setminus \zeta(s_i)} \|s_i - z_s\|_2$. The boundary ratio can be computed as:

$$r_i = 1 - \frac{d_i^I - d_i^N}{\max(d_i^N, d_i^I)}, \quad (19)$$

where a larger ratio $r_i$ indicates a higher probability that the instance lies near the decision boundary, and such boundary

samples are filtered to promote more robust learning. We adopt a linearly increasing ratio to control the threshold:

$$\gamma(e) = \gamma(e_0) + \frac{1 - \gamma(e_0)}{e - e_0} * (E - e_0), \quad (20)$$

where $e, e_0$ and $E$ denote the current, start and total TSC training step, respectively. Therefore, we rank the largest ratio among the synthesized time series data according to the boundary ratio $r_i$ of each instance, and progressively filter out the top-$\gamma(e)$ instances as the final augmented dataset $\tilde{\mathcal{D}}^{tr}$. Then time-series classifier is trained on $\mathcal{D}^{tr} = \mathcal{D}^{tr} \cup \tilde{\mathcal{D}}^{tr}$ for domain-generalized TSC.

### 3.5. Theoretical Analysis

To demonstrate how CURE works, we propose the following theoretical support. First, we elaborate the asymptotic behavior of $\mathcal{L}_{sup}$ for the analysis.

**Theorem 3.1.** *Assume time series with the same label are independent and identically distributed, and $|\mathcal{P}(i)| \to \infty, \forall i \in \{1, ..., N\}$ as $N \to \infty$, then with probability one we have*

$$\mathbb{E}\left[sim(c_i, c_l)|y_i = y_l\right] \geq \tau \limsup_{N \to \infty} \left(\frac{-\mathcal{L}_{sup}}{N}\right). \quad (21)$$

*What's more, if we consider cosine-similarity and further assume $\left\|\mathbb{E}\left[\frac{f_\psi(x_i)}{\|f_\psi(x_i)\|}\big|y_i\right]\right\|^2 = 1$, we have*

$$\limsup_{N \to \infty} \mathbb{E}\left[\frac{\mathcal{L}_{sup}}{N}\right] \leq -\frac{1}{\tau} + \Delta_\tau, \quad (22)$$

*where*

$$\Delta_\tau = \liminf_{N \to \infty} \frac{1}{N} \sum_{i=1}^N \log\left(\sum_{j \neq i} \mathbb{E} \exp\left(sim\left(c_i, c_j\right)/\tau\right)\right). \quad (23)$$

Eq. 21 indicates that by minimizing training loss $\mathcal{L}_{sup}$, we can improve the similarity of domain-invariant features of time series with the same label, which also provides a basis for the alignment of domain-invariant category information in the subsequent data generation process. During the adversarial tuning process, we aim to generate diverse time series data covering different domains while preserving label-guided domain-invariant information, the following theorem provides theoretical support for this.

**Theorem 3.2.** *Assume given domain $s \in \mathcal{S}$, $x_{1'}^0, ..., x_{N'}^0$ are $N$ time series generated for domain $s$ with independent and identically distributed, we have*

$$\sum_{i=1}^N \mathcal{L}_{dom}^i \geq \frac{1}{\tau|S|} \sum_{s=1}^{|S|} \sum_{i=1}^N \left[\sum_{s' \neq s} sim(f_\phi(x_i^0), z_{s'})\right.$$
$$\left. - (|S| - 1)sim(f_\phi(x_i^0), z_s)\right], \quad (24)$$

*Table 1.* Classification accuracy (%) with standard deviation ($\pm$) on four datasets. Models are trained on one source domain, and "T1–T4" denote target domains for each dataset. **Bold** values indicate the best results and underline values indicate the second best results.

| | Target | ANDMask | RSC | Mixup | DDLearn | AdaRNN | DIVERSIFY | PDEN | L2D | DI2SDiff | SEED | CURE |
|---|---|---|---|---|---|---|---|---|---|---|---|---|
| EMG | T1 | 44.79 ±5.78 | 53.47 ±2.52 | 52.43 ±1.56 | 55.38 ±2.29 | 57.47 ±1.42 | 55.38 ±2.29 | 52.17 ±3.87 | 56.60 ±1.25 | 58.24 ±3.41 | 76.04 ±3.78 | **77.02** ±1.50 |
| | T2 | 52.94 ±6.12 | 47.58 ±4.85 | 57.44 ±7.85 | 58.82 ±2.15 | 55.54 ±1.23 | 62.94 ±2.15 | 62.98 ±2.85 | 57.54 ±1.35 | 62.56 ±2.42 | 65.74 ±2.15 | **75.26** ±1.06 |
| | T3 | 66.52 ±2.54 | 60.92 ±1.68 | 67.66 ±5.85 | 77.19 ±3.11 | 68.85 ±2.54 | 73.16 ±3.11 | 78.88 ±4.78 | 74.62 ±2.65 | 79.89 ±1.87 | **82.22** ±1.58 | 80.78 ±2.03 |
| | Avg | 54.75 ±4.81 | 53.99 ±3.68 | 59.18 ±5.09 | 63.80 ±2.79 | 60.62 ±1.73 | 63.83 ±2.89 | 64.68 ±3.83 | 62.92 ±1.75 | 66.90 ±3.24 | 74.67 ±2.50 | **77.69** ±1.35 |
| DSADS | T1 | 44.12 ±4.56 | 51.45 ±4.78 | 45.37 ±4.87 | 55.79 ±1.27 | 52.25 ±5.12 | 54.82 ±1.27 | 48.63 ±4.65 | 55.10 ±1.20 | 54.16 ±2.55 | 60.81 ±1.74 | **71.39** ±1.37 |
| | T2 | 59.36 ±5.02 | 74.56 ±1.25 | 73.04 ±3.12 | 72.96 ±1.55 | 72.96 ±3.77 | 72.88 ±2.15 | 71.50 ±4.00 | 70.20 ±3.47 | 70.39 ±1.25 | 78.64 ±1.78 | **81.26** ±2.08 |
| | T3 | 47.66 ±5.77 | 55.61 ±4.65 | 60.03 ±2.71 | 50.37 ±0.78 | 56.21 ±5.10 | 60.91 ±0.78 | 51.92 ±2.12 | 56.21 ±0.87 | 61.03 ±0.74 | 67.18 ±1.55 | **70.04** ±1.49 |
| | Avg | 50.38 ±5.12 | 60.54 ±3.56 | 59.48 ±3.57 | 59.71 ±1.40 | 60.47 ±4.66 | 62.87 ±1.40 | 57.35 ±3.59 | 60.50 ±1.85 | 61.86 ±1.51 | 68.88 ±1.69 | **74.23** ±1.64 |
| PAMAP2 | T1 | 62.96 ±2.11 | 60.58 ±5.25 | 64.89 ±3.68 | 70.04 ±0.45 | 75.16 ±2.66 | 75.37 ±0.45 | 69.30 ±2.05 | 68.57 ±0.41 | 71.32 ±4.51 | 82.44 ±1.22 | **86.08** ±1.53 |
| | T2 | 62.57 ±2.14 | 60.44 ±3.55 | 54.87 ±3.25 | 70.97 ±1.25 | 69.47 ±4.12 | 72.04 ±1.25 | 55.75 ±2.15 | 52.75 ±6.41 | 73.27 ±2.40 | **77.19** ±1.78 | 74.74 ±1.42 |
| | T3 | 56.82 ±1.23 | 60.48 ±2.47 | 51.18 ±3.91 | 55.55 ±2.11 | 64.89 ±1.23 | 59.18 ±2.11 | 58.15 ±0.40 | 54.48 ±0.88 | 59.64 ±1.21 | 67.95 ±1.51 | **69.15** ±3.40 |
| | Avg | 60.78 ±1.83 | 60.50 ±3.76 | 56.98 ±3.61 | 65.52 ±1.27 | 69.84 ±2.34 | 68.86 ±1.27 | 61.07 ±1.53 | 58.60 ±2.57 | 68.08 ±2.71 | 75.86 ±1.50 | **76.66** ±2.12 |
| USC-HAD | T1 | 54.51 ±2.36 | 50.34 ±2.70 | 52.06 ±0.44 | 63.29 ±2.65 | 57.03 ±1.33 | 55.86 ±2.65 | 63.17 ±2.74 | 63.72 ±2.21 | 65.44 ±1.23 | 68.39 ±2.72 | **74.18** ±1.08 |
| | T2 | 57.49 ±2.12 | 57.42 ±5.78 | 76.88 ±4.87 | 79.53 ±2.15 | 66.01 ±1.23 | 80.23 ±2.15 | 74.79 ±0.25 | 73.54 ±2.33 | 74.58 ±4.97 | 83.13 ±1.03 | **83.66** ±1.70 |
| | T3 | 42.06 ±1.13 | 39.30 ±5.28 | 34.16 ±1.43 | 37.67 ±2.27 | 46.55 ±3.09 | 42.44 ±2.27 | 39.12 ±7.27 | 42.56 ±5.22 | 50.20 ±7.07 | **57.38** ±1.89 | 56.24 ±0.96 |
| | T4 | 49.62 ±8.92 | 44.98 ±4.23 | 53.36 ±0.77 | 40.19 ±8.21 | 56.53 ±0.83 | 51.90 ±8.21 | 45.52 ±4.82 | 55.71 ±1.80 | 49.18 ±3.77 | 63.06 ±1.20 | **64.57** ±5.42 |
| | Avg | 50.92 ±3.73 | 48.01 ±4.50 | 54.12 ±1.90 | 55.17 ±3.32 | 56.53 ±1.87 | 57.61 ±3.32 | 55.65 ±3.77 | 58.88 ±2.89 | 59.85 ±4.26 | 67.99 ±2.67 | **69.66** ±2.29 |
| Avg All | | 54.21 | 55.76 | 57.44 | 61.05 | 61.87 | 63.29 | 59.69 | 60.23 | 64.17 | 71.85 | **74.56** |

*if we further assume* $\mathbb{E}\left[sim(f_\phi(x_{i'}^0), f_\phi(x_{j'}^0))\right] = 1$, *then*

$$\liminf_{N \to \infty} \sum_{i=1}^N \mathcal{L}_{dom}^i \geq H_\tau - \frac{|\mathcal{S}| - 1}{\tau}, \qquad (25)$$

*where*

$$H_\tau = \frac{1}{\tau|\mathcal{S}|} \sum_{s=1}^{|\mathcal{S}|} \sum_{s' \neq s} \qquad (26)$$
$$\mathbb{E}\left[sim\left(f_\phi(x_{1'}^0), z_{s'}\right) \,\Big|\, domain(x_{1'}^0) = s\right].$$

Eq. 24 shows that the learning procedure of minimizing $\mathcal{L}_{dom}$ encourages the similarity of domain-specific features in the same domain while reducing the similarity in different domains, which helps us justify the differences between different domains. Detailed proofs for all theorems are provided in Appendix A.

## 4. Experiment

### 4.1. Experimental Setup

**Datasets.** To evaluate the effectiveness and robustness of our CURE, we conduct experiments on four widely used, publicly available cross-domain time series classification benchmarks following (Zhang et al., 2025): *EMG* (Senturk & Bakay, 2021), *DSADS* (Barshan & Altun, 2013), *PAMAP2* (Reiss, 2012), and *USC-HAD* (Zhang & Sawchuk, 2012). Details of the datasets are provided in Appendix B.

**Baselines.** We evaluate CURE against ten robust baselines from three major categories: (1) general-purpose OOD

methods including *ANDMask* (Parascandolo et al., 2020) and *RSC* (Huang et al., 2020); (2) classical SDG adaptations from computer vision including *Mixup* (Zhang et al., 2018), *PDEN* (Li et al., 2021), and *L2D* (Wang et al., 2021); and (3) state-of-the-art time-series DG techniques such as *AdaRNN* (Du et al., 2021), *DIVERSIFY* (Lu et al., 2022), *DDLearn* (Qin et al., 2023), *DI2SDiff* (Zhang et al., 2024), and *SEED* (Zhang et al., 2025).

**Implementation.** For our evaluation, we adopt a rigorous SDG protocol consistent with prior works in time series analysis (Lu et al., 2022; Qin et al., 2023). To ensure fair comparison with existing benchmarks, subjects within each dataset are first partitioned into predefined groups, and each group is treated as one domain. In the main setting reported in Table 1, we consider domain 0 as the source and treat all remaining domains as unseen targets. Our conditional generator is implemented as a 1D U-Net trained with the DDPM objective, using a base width of 64 and channel multipliers $(1, 2, 4, 8)$. The memory bank size is 4, with $\alpha = 0.8$ and $\beta = 1.0$, and we use 100 diffusion steps, 25 sampling steps, the AdamW optimizer with learning rate $2 \times 10^{-4}$, and a batch size of 32. In practice, the diffusion generator is trained once and then reused during progressive expansion, so the additional overhead mainly comes from offline sample generation and boundary-based filtering rather than repeated end-to-end retraining.

### 4.2. Performance Comparison

In our main experiments, we train models on domain 0 and evaluate on the remaining domains as targets. Table 1 summarizes the cross-domain classification accuracy on four

*Table 2.* **Detailed ablation study of classification accuracy (%)** on EMG and DSADS datasets. Models are optimized using one source domain and evaluated on target domains T1–T3. **Bold** values indicate the complete method performance.

| Variants | EMG | | | | DSADS | | | |
|---|---|---|---|---|---|---|---|---|
| | T1 | T2 | T3 | Avg | T1 | T2 | T3 | Avg |
| CURE w/o $\mathcal{L}_{ind}$ | $71.26_{\downarrow 5.76}$ | $69.37_{\downarrow 5.89}$ | $79.94_{\downarrow 0.84}$ | $73.53_{\downarrow 4.16}$ | $67.47_{\downarrow 3.92}$ | $74.55_{\downarrow 6.71}$ | $68.95_{\downarrow 1.09}$ | $70.32_{\downarrow 3.91}$ |
| CURE w/o $\mathcal{L}_{cat}$ | $63.92_{\downarrow 13.10}$ | $59.79_{\downarrow 15.47}$ | $65.27_{\downarrow 15.51}$ | $62.99_{\downarrow 14.70}$ | $66.85_{\downarrow 4.54}$ | $76.93_{\downarrow 4.33}$ | $62.96_{\downarrow 7.08}$ | $68.91_{\downarrow 5.32}$ |
| CURE w/o $\mathcal{L}_{dom}$ | $67.78_{\downarrow 9.24}$ | $64.18_{\downarrow 11.08}$ | $67.94_{\downarrow 12.84}$ | $66.63_{\downarrow 11.06}$ | $68.40_{\downarrow 2.99}$ | $74.15_{\downarrow 7.11}$ | $63.98_{\downarrow 6.06}$ | $68.84_{\downarrow 5.39}$ |
| CURE w/o Adversarial Tuning | $62.31_{\downarrow 14.71}$ | $59.88_{\downarrow 15.38}$ | $63.31_{\downarrow 17.47}$ | $61.83_{\downarrow 15.86}$ | $61.68_{\downarrow 9.71}$ | $80.50_{\uparrow 0.76}$ | $66.74_{\downarrow 3.30}$ | $69.64_{\downarrow 4.59}$ |
| CURE w/o Sample Filtering | $69.68_{\downarrow 7.34}$ | $70.81_{\downarrow 4.45}$ | $72.43_{\downarrow 8.35}$ | $70.97_{\downarrow 6.72}$ | $65.83_{\downarrow 5.56}$ | $82.38_{\uparrow 1.12}$ | $70.63_{\uparrow 0.59}$ | $72.95_{\downarrow 1.28}$ |
| **CURE** | **77.02** | **75.26** | **80.78** | **77.69** | **71.39** | **81.26** | **70.04** | **74.23** |

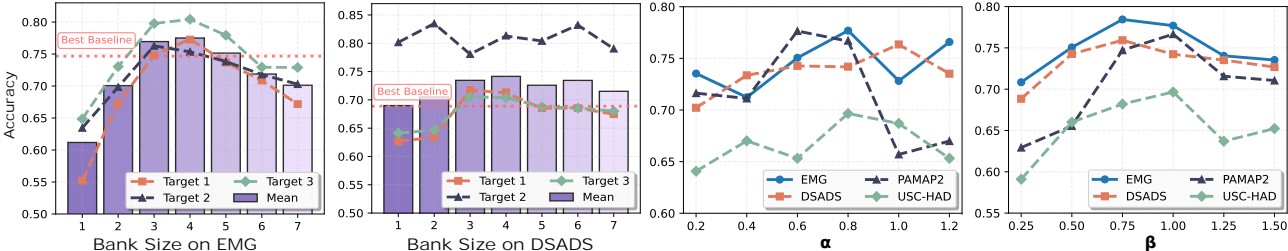

*Figure 2.* Parameter sensitivity analysis across different hyperparameter settings.

benchmarks under the single-domain generalization protocol. *First*, CURE consistently attains the strongest results across datasets and most target domains, showing substantial improvements over existing baselines particularly on challenging datasets (EMG, DSADS) while maintaining competitive performance on PAMAP2 and USC-HAD. This indicates robust generalization capabilities and broad applicability across different sensor modalities. *Second*, our approach benefits from explicitly modeling domain structure compared with augmentation-centric (Mixup, PDEN, L2D) and optimization-driven (AdaRNN, DIVERSIFY, DDLearn) baselines. Specifically, the feature disentanglement module extracts invariant semantics-aware information while preserving semantic-free domain contexts, while the diffusion-based adversarial tuning enriches sample diversity without drifting from class-discriminative regions. *Third*, the progressive diversity expansion scheme further stabilizes training and enhances generalization. Unlike other diffusion-based methods such as DI2SDiff, our approach incrementally updates the domain prompt memory bank and filters out boundary samples, which effectively broadens the coverage of diverse conditions while ensuring training stability.

### 4.3. Ablation Study

To evaluate the contribution of each component within CURE, we conduct a comprehensive ablation study using five variants: (1) *w/o* $\mathcal{L}_{ind}$, which removes the distance correlation constraint, allowing potential redundancy between invariant and domain-specific features; (2) *w/o* $\mathcal{L}_{cat}$, which discards the category semantic consistency loss, thus relaxing the alignment between generated and source samples;

(3) *w/o* $\mathcal{L}_{dom}$, which eliminates the domain style repulsion loss and the associated adversarial diversity mechanism; (4) *w/o Adversarial Tuning*, which disables the entire adversarial optimization process, including both consistency and diversity objectives; and (5) *w/o Sample Filtering*, which applies data augmentation to all generated samples without the progressive quality control of the boundary filtering mechanism. Table 2 summarizes the performance across these variants, and we have the following observations.

*First*, removing $\mathcal{L}_{cat}$ or $\mathcal{L}_{dom}$ causes the most pronounced degradation, particularly on the EMG dataset, confirming that explicit semantic guidance and adversarial domain exploration are indispensable for effective single-domain generalization. Without $\mathcal{L}_{cat}$, the diffusion model fails to preserve class-discriminative structures, while the absence of $\mathcal{L}_{dom}$ restricts the model's ability to explore diverse out-of-distribution styles. *Second*, the $\mathcal{L}_{ind}$ constraint ensures orthogonality between disentangled representations, and its removal causes a moderate performance decline, suggesting that preventing information leakage is beneficial for refining invariant features. *Third*, the ablation of adversarial tuning and sample filtering further validates our progressive enhancement strategy. The adversarial tuning mechanism successfully balances semantic consistency with style diversity, while the filtering mechanism proves vital for stabilizing training by removing unreliable boundary samples, with its impact being most pronounced on datasets with higher inherent complexity. This trend also suggests that the gains of CURE do not come from increasing synthetic diversity alone, but from retaining semantically reliable and mechanism-aware generated samples. Collectively, these

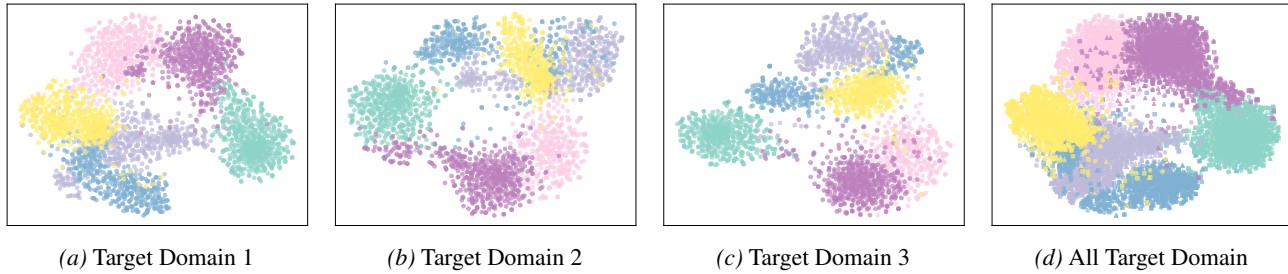

| *(a)* Target Domain 1 | *(b)* Target Domain 2 | *(c)* Target Domain 3 | *(d)* All Target Domain |

*Figure 3.* t-SNE visualizations of classification: feature distributions on three target domains and the overall distribution for EMG data.

findings demonstrate that the synergy of all components drives the superior generalization of CURE.

### 4.4. Parameter Sensitivity

We systematically analyze the sensitivity of our method to key hyperparameters—including memory bank size, trade-off parameter $\alpha$, and adversarial tuning parameter $\beta$. The results are illustrated in Figure 2.

► **Impact of Memory Bank Size.** We first analyze the impact of the memory bank size, which governs the diversity of domain prompts. As illustrated in Figure 2, increasing the number of prompts from very small bank sizes to a moderate range consistently improves performance, after which the curves fluctuate mildly or taper off. This trend confirms that incorporating richer semantic-free condition contexts facilitates broader distribution coverage and better generalization, while an overly large bank does not yield further gains. Moreover, CURE consistently outperforms strong baselines. Crucially, the performance remains high across a broad range of configurations, suggesting that our method is robust to bank size variations and does not rely on a narrowly tuned setting.

► **Impact of Loss Weights.** We further investigate the trade-off parameter $\alpha$ and adversarial tuning parameter $\beta$, which balance domain-specific feature learning and semantic consistency. Interestingly, both parameters exhibit a similar pattern of stability: while performance drops at extreme values, it remains consistently high within a broad optimal range (e.g., $\alpha \in [0.6, 0.8]$ and $\beta \in [0.75, 1.0]$). This meaningful overlap in optimal regions indicates that CURE maintains a stable equilibrium between semantic preservation and diversity expansion without requiring precise fine-tuning. Such insensitivity to hyperparameter variations implies that our method possesses strong transferability, allowing it to be easily adapted to new datasets.

### 4.5. Case Study

Figure 3a,3b, and 3c display the feature distributions for individual Target Domains 1, 2, and 3, respectively. In each subfigure, distinct clusters corresponding to different classes are clearly observable, indicating that within each specific target domain, the model effectively distinguishes between different classes, demonstrating strong intra-domain classification performance. Figure 3d presents the aggregated visualization samples from all target domains. The key observation is the remarkable coherence of class clusters across different domains, where samples belonging to the same class converge into unified clusters regardless of their original domain. This demonstrates that CURE effectively aligns the feature spaces of different target domains, enabling the model to learn invariant representations and achieve consistent classification performance across unseen domains.

## 5. Related Work

### 5.1. Time Series Classification.

Time Series Classification (TSC) assigns categorical labels to ordered sequences (Mohammadi Foumani et al., 2024), with applications spanning Human Activity Recognition (Chen et al., 2021; Miao et al., 2024; Wang et al., 2019) to healthcare diagnostics (Gong et al., 2024; Rajkomar et al., 2018; Schirrmeister et al., 2017). Deep learning approaches for TSC have evolved from Convolutional Neural Networks capturing local temporal patterns (Zhang et al., 2023a;b; Zheng et al., 2014; Ismail Fawaz et al., 2019) to recent architectures including Graph Neural Networks for multivariate dependencies (Wang et al., 2024b;a) and Transformers for long-range dependencies (Chowdhury et al., 2022; Gong et al., 2023; Wu et al., 2022a). Complementing these, contrastive learning aligns similar subsequences through various augmentation strategies (Eldele et al., 2021; Hyvarinen & Morioka, 2016; Tonekaboni et al., 2021; Yang et al., 2022; Yue et al., 2022). However, these methods primarily operate in self-supervised settings without addressing generalization to unseen domains.

### 5.2. Single Domain Generalization.

Domain generalization (DG) trains models robust to distributional shifts (Wang et al., 2022), either through domain-invariant feature learning (Ajakan et al., 2014; Erfani et al., 2016; Muandet et al., 2013) or explicit domain-specific

modeling (Mancini et al., 2018; Wang et al., 2020; Zhang et al., 2023c). Time series adaptations include identity cue exploitation (Qian et al., 2021), input augmentation (Qin et al., 2023), intra/inter-domain feature mixing (Zhang et al., 2024), shared physiological pattern extraction (Tu et al., 2024), and dynamic distribution modeling (Lu et al., 2022). Single-domain generalization (SDG) restricts training to one source domain (Li et al., 2021), typically employing raw-sequence transformations (Su et al., 2023), semantic interpolation (Xu et al., 2020; Zhang et al., 2018), or generative approaches (Guo et al., 2023; Wang et al., 2021), with subsequent improvements for realistic augmentations (Li et al., 2021; Yang et al., 2024). Beyond this line, metadata-aware conditional generation for time series (Narasimhan et al., 2024), text-conditioned diffusion (Zhou et al., 2025), and dynamic diffusion guidance (Koulischer et al., 2025) further suggest that richer control signals can improve sample diversity and controllability. Hard-sample construction via counterfactual negatives (Yang et al., 2023b) also highlights the importance of informative boundary cases for robust representation learning. While predominantly designed for images, existing time series DG methods (Lu et al., 2022; Qian et al., 2021; Qin et al., 2023; Sun & Saenko, 2016; Zhang et al., 2018; 2024; 2025) rely on multiple sources and degrade under SDG. Our framework addresses this gap by employing context-driven diffusion models to generate diverse, semantically consistent time series.

## 6. Conclusion

In this paper, we propose CURE, a prompt-driven diffusion framework for single-domain generalization in time series classification. By utilizing disentangled category and domain contexts as explicit prompts, CURE effectively addresses the diversity collapse challenge in single-domain training. Our progressive expansion strategy further enhances robustness by iteratively exploring diverse distributions via dynamic memory bank updates and boundary sample filtering. Extensive experiments confirm that CURE consistently outperforms state-of-the-art baselines, providing a novel and effective solution for generalizing from limited source data to diverse unseen target domains.

## Impact Statement

This paper presents CURE, a framework designed to improve single-domain generalization in time series classification. Our work advances the field of machine learning by addressing the challenge of model performance degradation when encountering unseen domains, particularly in sensor-based data. The potential societal consequences include more reliable and adaptable human activity recognition systems, which are foundational for applications in healthcare monitoring, physical rehabilitation, and mobile health technologies. By enabling models to generalize from a single source domain, our approach reduces the data collection and annotation burden for deploying AI solutions in new environments or for new individuals.

## Acknowledgements

Tao Ren is supported by the National Natural Science Foundation of China (62276058, 41774063), the Fundamental Research Funds for the Central Universities (N25GFZ011). Yifan Wang is supported by the Fundamental Research Funds for the Central Universities in UIBE (Grant No. 23QN02) and the Humanities and Social Sciences Research Fund of the Ministry of Education of China under Grant 25YJCZH275.

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

# A. Proof of Theoretical Support

## A.1. Proof of Theorem 3.1

Notice

$$\mathcal{L}_{sup} = -\sum_{i=1}^{N} \frac{1}{|\mathcal{P}(i)|} \sum_{l \in \mathcal{P}(i)} \log \frac{\exp\left(\text{sim}(c_i, c_l)/\tau\right)}{\sum_{j \neq i} \exp\left(\text{sim}(c_i, c_j)/\tau\right)}$$

$$= -\sum_{i=1}^{N} \frac{1}{|\mathcal{P}(i)|} \sum_{l \in \mathcal{P}(i)} \frac{\text{sim}(c_i, c_l)}{\tau} + \sum_{i=1}^{N} \log \left( \sum_{j \neq i} \exp\left(\text{sim}(c_i, c_j)/\tau\right) \right), \tag{27}$$

by law of large number, we have

$$\frac{1}{|\mathcal{P}(i)|} \sum_{l \in \mathcal{P}(i)} \frac{\text{sim}(c_i, c_l)}{\tau} \xrightarrow{a.s.} \frac{\mathbb{E}\left[\text{sim}(c_i, c_l)|y_i = y_l\right]}{\tau},$$

which implies $\mathbb{E}\left[\text{sim}(c_i, c_l)|y_i = y_l\right] \geq \tau \lim\sup_{N \to \infty} \left( \frac{-\mathcal{L}_{sup}}{N} \right)$. What's more, when using cosine-similarity and further assumptions we have

$$\mathbb{E}\left[\text{sim}(c_i, c_l)|y_i = y_l\right] = \mathbb{E}\left[ \frac{f_\psi\left(x_i\right)^\top f_\psi\left(x_l\right)}{||f_\psi\left(x_i\right)|| \cdot ||f_\psi\left(x_l\right)||} |y_i = y_l \right]$$

$$= \sum_{c=1}^{C} \mathbb{E}\left[ \frac{f_\psi\left(x_i\right)}{||f_\psi\left(x_i\right)||} |y_i = c \right]^\top \mathbb{E}\left[ \frac{f_\psi\left(x_l\right)}{||f_\psi\left(x_l\right)||} |y_l = c \right] \mathbb{P}\left(y_i = y_l = c\right)$$

$$= \sum_{c=1}^{C} \left\| \mathbb{E}\left[ \frac{f_\psi\left(x_i\right)}{||f_\psi\left(x_i\right)||} |y_i = c \right] \right\|^2 \mathbb{P}\left(y_i = y_l = c\right) \tag{28}$$

$$= \sum_{c=1}^{C} \mathbb{P}\left(y_i = y_l = c\right) = 1,$$

followed by Jensen's inequality we obtain

$$\lim\sup_{N \to \infty} \mathbb{E}\left[ \frac{\mathcal{L}_{sup}}{N} \right]$$

$$= -\frac{1}{N} \sum_{i=1}^{N} \frac{1}{\tau} + \lim\inf_{N \to \infty} \frac{1}{N} \sum_{i=1}^{N} \mathbb{E}\log \left( \sum_{j \neq i} \exp\left(\text{sim}(c_i, c_j)/\tau\right) \right)$$

$$\leq -\frac{1}{\tau} + \lim\inf_{N \to \infty} \frac{1}{N} \sum_{i=1}^{N} \log \left( \sum_{j \neq i} \mathbb{E}\exp\left(\text{sim}\left(c_i, c_j\right)/\tau\right) \right) \tag{29}$$

$$:= -\frac{1}{\tau} + \Delta_\tau.$$

■

### A.2. Proof of Theorem 3.2

Observe that given domain $s \in \mathcal{S}$, by continuity theorem

$$
\frac{1}{N} \sum_{i=1}^{N} \log \left( \exp \left( \mathrm{sim} \left( f_\phi(x_{i'}^0), z_s \right) / \tau \right) \right)
$$

$$
= \frac{1}{\tau N} \sum_{i=1}^{N} \mathrm{sim} \left( f_\phi(x_{i'}^0), z_s \right) \xrightarrow{a.s.} \frac{1}{\tau} \mathbb{E} \left[ \mathrm{sim} \left( f_\phi(x_{i'}^0), z_s \right) \right] \tag{30}
$$

$$
= \frac{1}{\tau N} \sum_{j=1}^{N} \mathbb{E} \left[ \mathrm{sim} \left( f_\phi(x_{i'}^0), f_\phi(x_{j'}^0) \right) \right] = 1/\tau,
$$

and by Jensen's inequality we have

$$
\log \left( \sum_{s'=1}^{|\mathcal{S}|} \exp \left( \mathrm{sim} \left( f_\phi(x_{i'}^0), z_{s'} \right) / \tau \right) \right)
$$

$$
= \log |\mathcal{S}| + \log \left( \frac{1}{|\mathcal{S}|} \sum_{s'=1}^{|\mathcal{S}|} \exp \left( \mathrm{sim} \left( f_\phi(x_{i'}^0), z_{s'} \right) / \tau \right) \right) \tag{31}
$$

$$
\geq \log |\mathcal{S}| + \frac{1}{|\mathcal{S}|} \sum_{s'=1}^{|\mathcal{S}|} \log \exp \left( \mathrm{sim} \left( f_\phi(x_{i'}^0), z_{s'} \right) / \tau \right)
$$

$$
= \log |\mathcal{S}| + \frac{\mathrm{sim} \left( f_\phi(x_{i'}^0), z_s \right)}{\tau |\mathcal{S}|} + \frac{1}{\tau |\mathcal{S}|} \sum_{s' \neq s} \mathrm{sim} \left( f_\phi(x_{i'}^0), z_{s'} \right),
$$

which further indicates

$$
\frac{1}{N|\mathcal{S}|} \sum_{i=1}^{N} \mathcal{L}_{dom}^i = -\frac{1}{N|\mathcal{S}|} \sum_{i=1}^{N} \sum_{s=1}^{|\mathcal{S}|} \log \frac{\exp \left( \mathrm{sim} \left( f_\phi(x_{i'}^0), z_s \right) \right)}{\sum_{s'=1}^{|\mathcal{S}|} \exp \left( \mathrm{sim} \left( f_\phi(x_{i'}^0), z_{s'} \right) / \tau \right)}
$$

$$
\geq -\frac{1}{\tau N|\mathcal{S}|} \sum_{s=1}^{|\mathcal{S}|} \sum_{i=1}^{N} \mathrm{sim} \left( f_\phi(x_{i'}^0), z_s \right) +
$$

$$
\frac{1}{N|\mathcal{S}|} \sum_{s=1}^{|\mathcal{S}|} \sum_{i=1}^{N} \left( \frac{\mathrm{sim} \left( f_\phi(x_{i'}^0), z_s \right)}{\tau |\mathcal{S}|} + \frac{1}{\tau |\mathcal{S}|} \sum_{s' \neq s} \mathrm{sim} \left( f_\phi(x_{i'}^0), z_{s'} \right) \right) \tag{32}
$$

$$
= \frac{1}{\tau N|\mathcal{S}|^2} \sum_{s=1}^{|\mathcal{S}|} \sum_{i=1}^{N} \left[ \sum_{s' \neq s} \mathrm{sim} \left( f_\phi(x_{i'}^0), z_{s'} \right) - (|\mathcal{S}| - 1) \mathrm{sim} \left( f_\phi(x_{i'}^0), z_s \right) \right],
$$

under further assumptions, by the law of large number we obtain

$$
\liminf_{N \to \infty} \frac{1}{N} \sum_{i=1}^{N} \mathcal{L}_{dom}^i
$$

$$
\geq \frac{1}{\tau |\mathcal{S}|} \sum_{s=1}^{|\mathcal{S}|} \mathbb{E} \left[ \sum_{s' \neq s} \mathrm{sim} \left( f_\phi(x_{1'}^0), z_{s'} \right) - (|\mathcal{S}| - 1) \mathrm{sim} \left( f_\phi(x_{1'}^0), z_s \right) \right] \tag{33}
$$

$$
:= H_\tau - \frac{|\mathcal{S}| - 1}{\tau},
$$

where $H_\tau = \frac{1}{\tau |\mathcal{S}|} \sum_{s=1}^{|\mathcal{S}|} \sum_{s' \neq s} \mathbb{E} \left[ \mathrm{sim} \left( f_\phi(x_{1'}^0), z_{s'} \right) | \mathrm{domain}(x_{1'}^0) = s \right].$ ∎

*Table 3.* Detailed statistics and preprocessing settings of the benchmark datasets.

| Dataset | # Subjects | # Classes | # Domains | Sensor Modality | Sampling Rate | Duration | Length | Final Shape |
|---|---|---|---|---|---|---|---|---|
| EMG | 36 | 6 | 4 | Electromyography | 200Hz | 1 second | 200 | $8 \times 1 \times 200$ |
| DSADS | 8 | 19 | 4 | IMU (5 positions) | 25Hz | 5 seconds | 125 | $45 \times 1 \times 125$ |
| PAMAP2 | 9 | 18 | 4 | IMU (3 positions) | 100Hz | 5.12 seconds | 512 | $27 \times 1 \times 512$ |
| USC-HAD | 14 | 12 | 5 | Motion Sensor | 100Hz | 5 seconds | 500 | $6 \times 1 \times 500$ |

## B. Dataset Details

In this work, we evaluate our proposed method on four publicly available time series classification datasets that are widely used in domain generalization research.

**EMG Dataset (Senturk & Bakay, 2021).** The EMG (Electromyography) dataset consists of electromyography signals recorded from 36 subjects performing six common hand gestures using a MYO Thalmic bracelet equipped with eight forearm sensors. The dataset is partitioned into 4 domains based on subject groups, with each subject performing 6 distinct hand gesture classes.

**DSADS Dataset (Barshan & Altun, 2013).** The Daily and Sports Activities Data Set (DSADS) offers a complex scenario with data from eight subjects across 19 physical activities. Recordings from five inertial measurement units (IMUs) positioned on different body parts introduce significant variability. The dataset is sampled at 25Hz and organized into 4 domains.

**PAMAP2 Dataset (Reiss, 2012).** The Physical Activity Monitoring Data Set 2 (PAMAP2) was gathered from nine participants performing 18 distinct physical activities, ranging from basic activities like walking and running to more complex exercises like rope jumping and nordic walking. The IMU data is sampled at 100Hz, while the heart rate data is sampled at approximately 9Hz. The dataset is divided into 4 domains based on subject groups.

**USC-HAD Dataset (Zhang & Sawchuk, 2012).** The University of Southern California Human Activity Dataset (USC-HAD) focuses on 12 core activities captured from 14 subjects using a single waist-mounted motion sensor. The dataset is sampled at 100Hz and organized into 5 domains.

*Table 4.* Classification accuracy (%) on four datasets, each trained on a source domain and tested on the remaining domains.

| | Source | ANDMask | RSC | Mixup | DDLearn | AdaRNN | DIVERSIFY | PDEN | L2D | DI2SDiff | SEED | CURE |
|---|---|---|---|---|---|---|---|---|---|---|---|---|
| EMG | 0 | 54.75 | 53.99 | 59.18 | 63.80 | 60.62 | 63.83 | 64.68 | 62.92 | 66.90 | 74.67 | 77.69 |
| | 1 | 59.92 | 45.90 | 62.22 | 66.58 | 58.89 | 62.62 | 68.27 | 61.29 | 64.71 | 70.79 | 77.37 |
| | 2 | 45.14 | 57.78 | 65.65 | 64.17 | 58.75 | 62.50 | 62.70 | 63.96 | 65.82 | 69.42 | 75.85 |
| | 3 | 59.81 | 49.73 | 52.62 | 61.44 | 69.86 | 70.91 | 68.17 | 63.22 | 68.54 | 74.83 | 75.52 |
| | Avg | 54.91 | 51.85 | 59.92 | 64.00 | 62.03 | 64.96 | 65.95 | 62.85 | 66.49 | 72.43 | 76.61 |
| DSADS | 0 | 50.38 | 60.54 | 59.48 | 59.71 | 60.47 | 62.87 | 57.35 | 60.50 | 61.86 | 68.88 | 74.23 |
| | 1 | 60.68 | 61.94 | 58.70 | 54.59 | 65.10 | 62.71 | 57.47 | 60.27 | 60.54 | 67.30 | 75.26 |
| | 2 | 43.35 | 50.82 | 51.97 | 58.63 | 55.62 | 60.44 | 55.82 | 60.44 | 60.58 | 68.96 | 67.10 |
| | 3 | 36.19 | 67.65 | 65.67 | 55.18 | 61.42 | 60.68 | 55.98 | 59.87 | 65.80 | 71.03 | 75.07 |
| | Avg | 47.65 | 60.24 | 58.96 | 57.03 | 60.65 | 61.68 | 56.66 | 60.27 | 62.20 | 69.04 | 72.92 |
| PAMAP2 | 0 | 60.78 | 60.50 | 56.98 | 65.52 | 69.84 | 68.86 | 61.07 | 58.60 | 68.08 | 75.86 | 76.66 |
| | 1 | 53.33 | 59.43 | 52.15 | 64.85 | 67.38 | 70.54 | 62.71 | 51.67 | 68.51 | 72.54 | 74.63 |
| | 2 | 62.57 | 61.60 | 55.14 | 68.70 | 71.65 | 69.06 | 58.27 | 60.99 | 69.87 | 76.83 | 77.51 |
| | 3 | 62.82 | 69.73 | 61.22 | 69.34 | 72.97 | 68.52 | 59.77 | 61.57 | 71.52 | 76.71 | 76.82 |
| | Avg | 59.88 | 62.82 | 56.37 | 67.10 | 70.46 | 69.25 | 60.45 | 58.21 | 69.49 | 75.49 | 76.41 |
| USC-HAD | 0 | 50.92 | 48.01 | 54.12 | 55.17 | 56.53 | 57.61 | 55.65 | 58.88 | 59.85 | 67.99 | 69.66 |
| | 1 | 50.13 | 44.37 | 57.36 | 43.19 | 58.19 | 57.58 | 59.20 | 54.07 | 56.51 | 67.83 | 69.29 |
| | 2 | 49.54 | 48.93 | 58.24 | 58.26 | 52.56 | 54.95 | 55.40 | 59.55 | 60.54 | 67.02 | 68.87 |
| | 3 | 49.81 | 51.70 | 51.82 | 63.89 | 54.27 | 55.31 | 53.50 | 64.11 | 64.85 | 68.89 | 65.51 |
| | Avg | 50.10 | 48.25 | 55.38 | 55.13 | 55.39 | 56.36 | 55.94 | 59.15 | 60.44 | 67.93 | 68.33 |
| | Avg All | 53.13 | 55.79 | 57.66 | 60.81 | 62.13 | 63.06 | 59.75 | 60.12 | 64.65 | 71.22 | 73.57 |

*Table 5.* Ablation study of classification accuracy (%) on four datasets.

| Variants | EMG | DSADS | PAMAP2 | USC-HAD | Avg All |
|---|---|---|---|---|---|
| CURE | **77.69** | **74.23** | **76.66** | **69.66** | **74.56** |
| CURE w/o $\mathcal{L}_{ind}$ | $73.53_{\downarrow 4.16}$ | $70.32_{\downarrow 3.91}$ | $72.61_{\downarrow 4.05}$ | $65.23_{\downarrow 4.43}$ | $70.42_{\downarrow 4.14}$ |
| CURE w/o $\mathcal{L}_{cat}$ | $62.99_{\downarrow 14.70}$ | $68.91_{\downarrow 5.32}$ | $62.87_{\downarrow 13.79}$ | $37.26_{\downarrow 32.40}$ | $58.01_{\downarrow 16.55}$ |
| CURE w/o $\mathcal{L}_{dom}$ | $66.63_{\downarrow 11.06}$ | $68.84_{\downarrow 5.39}$ | $62.93_{\downarrow 13.73}$ | $59.10_{\downarrow 10.56}$ | $64.38_{\downarrow 10.18}$ |
| CURE w/o Adversarial Tuning | $61.83_{\downarrow 15.86}$ | $69.64_{\downarrow 4.59}$ | $72.68_{\downarrow 3.98}$ | $63.17_{\downarrow 6.49}$ | $66.83_{\downarrow 7.73}$ |
| CURE w/o Sample Filtering | $70.97_{\downarrow 6.72}$ | $72.95_{\downarrow 1.28}$ | $64.28_{\downarrow 12.38}$ | $58.79_{\downarrow 10.87}$ | $66.75_{\downarrow 7.81}$ |

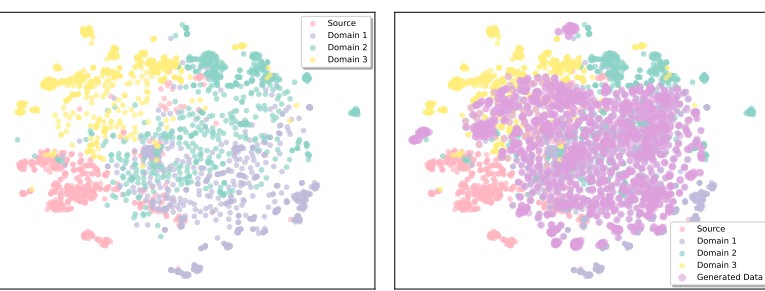

*Figure 4.* t-SNE visualizations of domain distributions on PAMAP dataset. Left: Source and target domains. Right: Generated domains (marked in purple) overlaid on the original domain distributions.

Table 3 summarizes the main features of the adopted datasets, including the number of subjects, activity classes, domains, and sensor modalities. To ensure consistency and reproducibility across all datasets, we apply standardized preprocessing procedures. The preprocessing pipeline includes data segmentation, normalization, and format transformation to prepare the raw sensor data for model training and evaluation. We utilize sliding windows to segment the continuous time series data into fixed-length samples. The window parameters follow established practices in the literature (Barshan & Altun, 2013). For the EMG dataset, we normalize the data using MinMaxScaler to ensure that all features are scaled to the range [0, 1]. For the other datasets (DSADS, PAMAP2, and USC-HAD), standard normalization is applied to the raw sensor data. The detailed preprocessing settings for all datasets are summarized in Table 3, which provides comprehensive information for reproducibility and comparison with other methods.

## C. More Experimental Results

**More Ablation Studies** We provide additional ablation studies across all four datasets (EMG, DSADS, PAMAP2, and USC-HAD) to complement the detailed analysis presented in the main paper. The comprehensive results shown in Table 5 confirm that the category semantic consistency loss ($\mathcal{L}_{cat}$) and the domain style repulsion loss ($\mathcal{L}_{dom}$) are the most influential individual objectives overall, while the adversarial tuning mechanism consistently contributes to improved performance. The progressive learning strategy demonstrates effectiveness across different dataset characteristics, validating the robustness of our approach.

**All Single-Domain Generalization Results** This section presents comprehensive experimental results for single-domain generalization across all four datasets. We evaluate our proposed method against ten baseline approaches using a rigorous leave-one-domain-out evaluation protocol. For each dataset, we train models on one source domain and test on the remaining target domains, reporting the average performance across all possible domain splits.

Table 4 shows the detailed classification accuracy results for each dataset. Our method consistently outperforms all baseline approaches across different datasets, demonstrating its effectiveness in handling domain shift challenges. The results indicate that our approach achieves superior generalization capabilities by leveraging the proposed domain-aware diffusion framework and progressive boundary sample filtering strategy.

**Computational Cost** To clarify the practical overhead of CURE, we report its stage-wise runtime on the EMG benchmark. The representation disentanglement module requires 5.55 seconds per epoch, and the conditional diffusion generator requires 8.59 seconds per epoch during its one-off training stage. Afterward, the trained generator is reused during progressive

*Table 6.* Comparison with alternative generative augmentation strategies on the EMG benchmark.

| Method | EMG Avg. Acc. (%) |
|--------|-------------------|
| CURE | **77.69** |
| GAN | 72.68 |
| VAE | 69.37 |
| Latent interpolation | 61.56 |

expansion, where the full offline sample generation and boundary-based filtering process takes 103.95 seconds in total. The final classifier retraining stage costs 1.55 seconds per epoch. Therefore, the main extra cost of CURE lies in the one-time preparation and offline synthesis stages, while the online classifier update remains lightweight once the generator has been obtained.

**Generative Augmentation Comparison** To further justify the choice of diffusion-based generation, we compare CURE with several simpler generative augmentation alternatives on the EMG benchmark under the same downstream classifier and training protocol. As shown in Table 6, the diffusion-based design achieves the strongest accuracy among these options. This result suggests that, for the current task, diffusion is better suited to preserving class semantics while enabling controllable style variation during augmentation.

**Domain Visualization** To demonstrate the effectiveness of our progressive diversity expansion framework in addressing the limited diversity challenge in single domain-generalized time series classification, we visualize the domain distribution evolution using t-SNE embeddings on the PAMAP dataset. To examine the domain distribution characteristics, we first train a domain classification model using the original domains and then utilize the learned features from this model for visualization purposes. Figure 4 presents a compelling comparison between the original domain distributions and the generated domain distributions after applying our CURE framework. In the original domains visualization, we observe that the four distinct domains exhibit relatively separated clustering patterns. The generated domains visualization demonstrates a remarkable transformation in the domain distribution landscape. The most striking observation is the emergence of a large, dense cluster of generated data that effectively encompasses and unifies the target domains. This visualization validates our core hypothesis that explicit semantic guidance through disentangled category and domain context can effectively address the limited diversity problem in single-domain generalization.

