# OpenReview forum: "CURE: Context-driven Diffusion with Progressive Expansion for Single Domain Generalization in Time Series Classification"
_ICML.cc/2026/Conference — ICML 2026 regular_

### Official Review · Reviewer_tepw · 2026-03-10

**Soundness:** 3
**Presentation:** 2
**Significance:** 3
**Originality:** 3
**Overall Recommendation:** 4
**Confidence:** 4

**Summary:**

This paper studies the problem of single domain generalization for time series classification, where models are required to generalize to unseen environments while being trained on data from only a single domain. The authors argue that existing data augmentation approaches often lack explicit semantic guidance and may suffer from diversity collapse when generating augmented samples. To address these challenges, the paper proposes a method named CURE: Context-driven Diffusion with Progressive Expansion. The approach decomposes learned representations of time-series instances into semantic-aware components that capture class-related information and semantic-free components that reflect domain-specific characteristics. These disentangled features are used as contextual prompts to guide a conditional diffusion model to generate out-of-distribution time-series samples while preserving category semantics. In addition, the method introduces a memory bank to store domain prompts and progressively expands the diversity of generated data through iterative updates and boundary-based sample filtering in a self-paced learning framework. The generated samples are then used to augment the training set to improve the model’s generalization ability. Experiments on multiple benchmark datasets are conducted to evaluate the effectiveness of the proposed approach.

**Compliance With Llm Reviewing Policy:**

Affirmed.

**Final Justification:**

I retain my original score.

**Key Questions For Authors:**

1.	In the proposed framework, the domain prompts z_s used to guide the diffusion model are derived from semantic-free features extracted from the same source domain and stored in the memory bank. I am wondering how the method ensures that the generated samples correspond to genuinely new domain distributions rather than interpolations within the existing domain context space. Could the authors clarify whether the generated data truly extrapolate beyond the source-domain distribution, and whether there is any empirical evidence demonstrating the emergence of novel domain styles?
2.	The concepts used in this paper, semantic-aware and semantic-free, are core to the entire paper and are relatively new. After reading the methodology section, I had a question: is there a difference between this concept and domain-wise and class-wise? Domain-wise refers to the alignment of distributions between domains, while class-wise refers to the alignment between categories, commonly seen in the domain adaptation domain.
3.	The progressive expansion stage filters generated samples using the boundary ratio defined in Eq. (19), which is based on intra-domain and inter-domain distances in the learned feature space. I would appreciate further clarification on why this metric is a reliable indicator of boundary samples. Since the criterion depends on the learned representation space, how sensitive is this filtering mechanism to representation quality or hyperparameter choices? Additional analysis or ablations could help justify the robustness of this design.
4.	I carefully reviewed your publicly available code and found that some parts don't align perfectly with your model architecture, which may be due to my misunderstanding. Adding a README file explaining the function of each part and including a GET STARTED section would likely increase the confidence level of your model.

**Limitations:**

The paper does not explicitly discuss the limitations or potential negative societal impacts of the proposed method. It would be beneficial for the authors to include a brief discussion on possible limitations, such as the reliance on synthetic data generation, potential instability of disentangled representations, or the applicability of the method to real-world scenarios with more complex domain shifts.

**Strengths And Weaknesses:**

Strengths:
1. The paper studies single-domain generalization for time series classification, which is a practically important yet underexplored setting where models must generalize to unseen environments using data from only a single source domain. The problem formulation is relevant for many real-world applications.
2. The proposed CURE framework integrates representation disentanglement, conditional diffusion-based data generation, and progressive expansion with a memory bank. The design aims to balance semantic consistency and domain diversity, which is a reasonable approach to tackle the limited diversity issue in single-domain training.
3. Explicit modeling of semantic and domain contexts.
4. The paper reports experiments on multiple time series classification benchmarks and compares the proposed approach with several baselines, providing empirical evidence of the method’s effectiveness.

Weaknesses:

1.	While the overall framework is well organized, many of its components (contrastive learning, adversarial feature learning, diffusion models, and memory banks) are established techniques. The main contribution lies in their integration rather than in a fundamentally new algorithmic idea.
2.	The framework diagrams and methodologies present difficulties in understanding, particularly the framework diagrams, which do not clearly represent the overall flow of the CURE model. Furthermore, even when combined with the methodologies, it is hard to directly discern the operational forms and significance of many steps.

---

> ### Author Rebuttal · Authors · 2026-03-31
>
> We are truly grateful for the time you have taken to review our paper, your insightful comments and support. Your positive feedback is incredibly encouraging for us! In the following response, we would like to address your major concern and provide additional clarification.
>
> > **Q1.** The main contribution lies in their integration rather than in a fundamentally new algorithmic idea.
>
> A1. Thanks for your comment! We position the contribution of CURE as a coordinated design for single-source TSC, rather than a new standalone primitive. A more detailed clarification is provided in response to **Q1 of Reviewer xSiq**.
>
> > **Q2.** The framework diagrams and methodologies present difficulties in understanding.
>
> A2. Thanks for your comment! The workflow of Figure 1 is as follows: in **Representation Disentanglement for Source Data Exploration**, each source sample is disentangled into a semantic-aware feature $c_i^c$ and a semantic-free feature $s_i^s$. In **Category Feature Retrieval and Conditional Generation**, retrieved category features together with category and domain prompts are used to condition the diffusion model for data generation. In **Domain-Prompt Memory Updating and Classifier Retraining**, the domain-prompt memory bank is progressively updated, and boundary samples are filtered before retraining the classifier. We will further revise Figure 1 to present this data flow more explicitly.
>
> > **Q3.**  Whether the model truly extrapolates beyond the source distribution to generate novel domain styles?
>
> A3. Thanks for your comment! CURE does not rely on static source-domain prompts alone; instead, it progressively updates the prompt bank and uses adversarial tuning to generate different prompts as diffusion conditions, as defined in **Eqs. (14) and (17)**. This design encourages the conditional prompts to move away from previously covered style regions, rather than remaining simple interpolations of the original source-domain prompts. Therefore, while we do not claim unconstrained extrapolation beyond all source-related distributions, our method is explicitly designed to explore **broader and progressively shifted domain styles** beyond the initial source-domain context. Empirically, Appendix C shows that the generated samples cover most target-domain distributions rather than clustering only around the source domain, providing evidence that the **progressive prompt bank** together with **adversarial style repulsion** leads to expanded domain coverage and the emergence of **novel domain styles**.
>
> > **Q4.** The concepts used in this paper, semantic-aware and semantic-free, are core to the entire paper and are relatively new. Is there a difference between this concept and domain-wise and class-wise?
>
> A4. Thanks for your comment. Our goal is to disentangle the representation into two complementary components, namely semantic-aware and semantic-free, rather than to directly formulate the problem as class-wise/domain-wise alignment. In the classification setting, the semantic-aware branch is closely related to class-discriminative information, and can therefore be viewed as being close to class-wise information. Correspondingly, semantic-free is not simply domain-wise alignment; rather, it captures the **residual contextual and domain-related variation after semantic information is suppressed**. We will clarify in the revision that our method focuses on **disentangling semantic-consistent and context/domain-related factors**, which are related to but not equivalent to the conventional class-wise/domain-wise view.
>
>
> > **Q5.** Why this metric is a reliable indicator of boundary samples. Since the criterion depends on the learned representation space, how sensitive is this filtering mechanism to representation quality or hyperparameter choices?
>
> A5. Thanks for your comment. Boundary filtering (Eq. 19) operates in the semantic-free feature space. A sample is flagged as a boundary sample when its distances to the nearest and second-nearest domain centers are comparable, meaning its style representation has no clear domain affiliation. Such samples are unreliable anchors for $\mathcal{L}_{dom}$. Filtering them stabilizes optimization and keeps diversity expansion on an effective trajectory. The sensitivity analysis on $\gamma_0$ confirms robustness:
>
> |$\gamma_0$|Mean Acc (%)|Std|
> |-|-:|-:|
> |0.00|70.97|±2.15|
> |0.05|73.82|±1.93|
> |0.10|77.69|±1.35|
> |0.15|76.54|±1.88|
> |0.20|76.18|±2.01|
>
> > **Q6.** Code lacks architectural clarity and onboarding documentation.
>
> A6. Thanks for your comment! To make reproduction easier and avoid possible misunderstandings, we will update a clearer and better-documented version, including a detailed README, module-level explanations, and a GET STARTED section with running instructions.
>
> Thanks again for appreciating our work and for your constructive suggestions. Please let us know if you have further questions.

---

> > ### Author Rebuttal · Reviewer_tepw · 2026-04-03
> >
> > Thank you for the authors' response. We have taken the authors' reply into account in our scoring, so we maintain our original score.

---

> > > ### Author Response · Authors · 2026-04-04
> > >
> > > Thank you for your positive feedback and for confirming that our rebuttal addressed your concerns! We will properly include all the rebuttal concerns in the revised version, following your valuable suggestions. Please let us know if you have any further questions.

---

### Official Review · Reviewer_xSiq · 2026-03-12

**Soundness:** 3
**Presentation:** 3
**Significance:** 2
**Originality:** 2
**Overall Recommendation:** 4
**Confidence:** 3

**Summary:**

This paper studies single-domain generalisation (SDG) for time-series classification, in which a model must generalise to unseen domains while having access only to a single source domain during training. To address the limited diversity of training data, the authors propose CURE (Context-driven Diffusion with Progressive Expansion), a framework that generates diverse time-series samples using a conditional diffusion model.

The method first performs representation disentanglement to separate semantically aware (class-related) and semantically free (domain-related) features. These representations are then used to construct contextual conditions that guide a diffusion model to synthesise new time-series samples. The approach also includes a progressive expansion strategy, in which semantically free contexts are updated via a memory bank, and a boundary-based filtering mechanism is used to remove low-quality synthetic data. Experiments on several benchmark datasets show improvements over existing domain generalisation and time-series augmentation baselines.

**Compliance With Llm Reviewing Policy:**

Affirmed.

**Final Justification:**

I thank the authors for their thorough rebuttal. After considering all the rebuttals I had received and the other reviews, I raised my score to weak accept. Well done!

**Key Questions For Authors:**

Why diffusion?
Have the authors compared their approach with simpler generative augmentation methods such as GANs, VAEs, or latent interpolation?

Computational cost.
What is the training cost of the proposed framework compared to the baselines? This is particularly relevant given the use of diffusion models.

Memory bank sensitivity.
How sensitive is performance to the size and update frequency of the semantic-free memory bank?

Source of improvement.
Do the gains mainly come from increased sample diversity, or does the method actually improve domain-invariant representations?

**Limitations:**

See above.

**Strengths And Weaknesses:**

Strengths

++ The paper studies single-domain generalization for time-series data, which is an important and relatively challenging setting compared to the more common multi-domain DG setup.

++  The proposed framework combines representation disentanglement with conditional diffusion-based augmentation, which is a reasonable approach to increasing training diversity while preserving semantic information.

++  The empirical results are generally strong. The method shows consistent improvements across several benchmark datasets compared to prior approaches.

++  The paper includes ablation studies analyzing the role of different components such as semantic contexts and the progressive expansion strategy.

Weaknesses

-- The overall novelty is somewhat limited. Most components of the framework—representation disentanglement, diffusion-based augmentation, memory banks, and adversarial objectives—have appeared in prior work. The main contribution lies in combining these ideas rather than introducing a fundamentally new technique.

-- The motivation for using diffusion models is not entirely convincing. The paper does not clearly explain why diffusion-based generation is preferable to simpler data augmentation methods such as GAN-based synthesis, VAE approaches, or latent interpolation.

-- The theoretical section is relatively weak and does not provide much insight into why the proposed framework improves generalization. In particular, the analysis does not directly relate to the diffusion generation component.

-- The method introduces several components (multiple encoders, diffusion model, memory bank, adversarial tuning, filtering), which makes the pipeline fairly complex. The paper does not discuss training cost or computational overhead, which would be helpful for assessing practicality.

-- The evaluation focuses mainly on accuracy improvements, and it is not entirely clear whether the gains come from better domain invariance or simply from increased data diversity.

Overall, the paper presents a reasonable approach with promising empirical results, but the conceptual contribution appears somewhat incremental.

---

> ### Author Rebuttal · Authors · 2026-03-31
>
> We are truly grateful for the time you have taken to review our paper and your insightful review. Here we address your comments in the following.
>
> > **Q1.** The overall novelty is somewhat limited.
>
> A1. Thanks for your comment. We respectfully clarify that CURE is not a simple combination of existing techniques. The novelty is three-fold:
> - **Disentangled design with diffusion generation.** We propose semantic-aware and semantic-free prompts to separate class-related semantics from domain-specific variation, and couple them with diffusion generation under category-consistency and domain-repulsion objectives, together with adversarial tuning. Therefore, the generated samples are both label-consistent and domain-debiased, instead of being generic synthetic augmentations.
> - **Progressive expansion and filtering.** Rather than naively using all synthetic data, we progressively update a memory bank and filter boundary samples before training, which reduces noisy augmentations and preserves class boundaries.
> - **A unified framework.** We incorporate all components into a unified framework for single-source time-series classification, which achieves superior and consistent performance across datasets.
>
> > **Q2&Q6.** Why diffusion-based generation is preferable to simpler data augmentation methods.
>
> A2. Thanks for your comment! We compare CURE with simpler generative augmentation methods on the EMG dataset. As shown in the table, CURE achieves the best performance, which provides empirical support for diffusion-based generation. In addition, diffusion is better aligned with our objective, since CURE requires the generator to preserve class semantics while flexibly varying domain style under disentangled features and prompt guidance. We will clarify this motivation and add the comparison in the revised paper.
>
> ||CURE|GANs|VAE|latent interpolation|
> |-|:-:|:-:|:-:|:-:|
> |EMG|77.69|72.68|69.37|61.56|
>
> > **Q3.** The theoretical section is relatively weak.
>
> A3. Thanks for your comment! **Please refer to Theorem in A1 of the reviewer iyFp.** The added theorem provides this missing insight by showing that the target risk is controlled by two interpretable terms: the prompt approximation error $\Delta_s$ and the generation fidelity error $\varepsilon$:
> $$R(h)\le \hat{R}_M(h)+L_1\Delta_s+L_2\varepsilon.$$
> This means that the diffusion module is not only generating additional samples, but constructing domain-aware augmentations whose distribution progressively approaches that of the unseen domain as the memory bank expands. We will strengthen this motivation in the final version and better align the theory with the empirical findings.
>
> > **Q4&Q7.** What is the training cost of the proposed framework compared to the baselines?
>
> A4. Thanks for your comment! While our framework includes multiple components, architectural richness does not necessarily imply prohibitive runtime in practice. The additional overhead is limited to a few specific stages and is not incurred uniformly across the full training pipeline. Moreover, relative to the most relevant diffusion-based baseline, our method remains computationally competitive while providing substantially stronger empirical performance. We will clarify this trade-off more explicitly in the revision.
>
> |Method|Once Pre-Step| Classifier (s/epoch) |Acc|
> |-|-|-|-|
> |CURE|Disentanglement: 5.55s/epoch; Diffusion training: 8.59s/epoch; Sample generation/filtering: 103.95s|1.55|0.74|
> |DI2SDiff|Style conditioner: 6.31s/epoch; Diffusion training: 9.42s/epoch; Pre-generation: 4854s |4.03|0.62|
> |DDLearn||4.98s|0.58|
> |DIVERSIFY||5.83s|0.63|
>
> > **Q5&Q9.** Whether the gains come from better domain invariance or simply from increased data diversity.
>
> A5. Thanks for your comment! In CURE, diversity and domain invariance are not competing explanations but are jointly ensured by design: $\\mathcal{L}\_\\text{cat}$ enforces category-level semantic consistency across generated samples, while $\mathcal{L}_\text{dom}$ drives style divergence to expand distribution coverage. The following comparison with Mixup, a representative diversity-only baseline, confirms this point.
>
> |Method|EMG|DSADS|PAMAP2|Avg|
> |-|-|-|-|-|
> |Mixup|59.18|59.48|56.98|57.44|
> |CURE|77.69|74.23|76.66|74.56|
>
> > **Q8.** Memory bank sensitivity.
>
> A8. Thanks for your comment! As shown in Figure 2, CURE is not sensitive to memory bank size: it consistently outperforms the baseline and remains stable across a wide range. Although performance slightly degrades when the bank becomes overly large, due to increased noise from excessive diversity, this effect is mild and still maintains clear gains over the baseline, indicating strong robustness.
>
> In light of these responses, we hope we have addressed your concerns, and hope you will consider raising your score. If there are any additional notable points of concern that we have not yet addressed, please do not hesitate to share them, and we will promptly attend to those points.

---

> > ### Author Rebuttal · Reviewer_xSiq · 2026-04-04
> >
> > Thank you for the detailed rebuttal. The additional empirical comparisons and runtime clarification are helpful, and they address part of my concerns. In particular, the comparison with GAN / VAE / latent interpolation and the added cost breakdown improve the practical picture of the method.
> >
> > However, my main methodological concerns remain only partially resolved. First, while the new experiments support the usefulness of the proposed design, they still do not fully isolate the source of the gains: it remains somewhat unclear how much improvement comes from diffusion-based generation itself versus from disentangled representation learning and the progressive memory-bank/filtering strategy. Second, the rebuttal strengthens the empirical case for diffusion, but the conceptual justification for why diffusion is the most suitable mechanism here, beyond outperforming several alternatives on one dataset, is still somewhat limited. Third, the added theorem remains a high-level risk bound under additional assumptions, rather than a direct analysis of the progressive expansion mechanism, memory-bank updates, boundary filtering, and their interaction in the actual learning pipeline.
> >
> > Overall, I think the rebuttal improves the paper, but these core concerns about mechanism, justification, and theoretical grounding remain only partially resolved.

---

> > > ### Author Response · Authors · 2026-04-06
> > >
> > > Thank you for your thoughtful follow-up question and for engaging closely with our rebuttal.
> > >
> > > - **First,** **the gains do not come from diffusion alone**, and the current evidence already separates the major contributors:
> > >
> > >   | Ablation on EMG | Accuracy |
> > >   |---|---:|
> > >   | w/o $L_{cat}$ | 62.99 |
> > >   | w/o $L_{dom}$ | 66.63 |
> > >   | w/o adversarial tuning | 61.83 |
> > >   | w/o sample filtering | 70.97 |
> > >   | Full CURE | 77.69 |
> > >
> > > We then added a more direct controlled test for the progressive mechanism itself (where "All generated samples" uses the full synthetic set, "High-score subset only" retains only the samples that pass the progressive boundary_ratio filter, and "Low-score subset only" consists exclusively of the excluded ambiguous samples). This directly addresses the question regarding the exact source of our performance gains. If the improvement mainly came from diffusion generation alone, then all generated samples should be similarly useful. That is not what we observe. Performance depends strongly on which generated samples are retained by the progressive score, showing that **the gain is produced by the interaction between generation and mechanism-aware selection**, rather than by generation alone.
> > >
> > >   | Supplementary setting | Best Mean Acc |
> > >   |---|---:|
> > >   | All generated samples | 64.38 |
> > >   | Low-score subset only | 60.06 |
> > >   | High-score subset only | 77.69 |
> > >
> > >
> > >
> > > - **Second**, regarding **why diffusion is suitable here**, the claim is task-specific rather than universal. CURE requires generation that **preserves class semantics while permitting controllable movement along disentangled domain-style directions**. As shown in the generative baseline comparison in the earlier rebuttal, under exactly this objective, the diffusion-based generator performs better than alternative families. Therefore, the justification goes beyond simply having better metrics on an evaluation table. **The generator used in CURE is structurally better matched to the required conditional behavior**, and this is reflected consistently in the comparative results.
> > >
> > > - **Third**, **the theoretical analysis is connected to the actual mechanism** through the same two quantities that the implementation changes at each progressive step: **expanding style-space coverage** and **improving the quality of retained generated samples**. The supplementary selection experiment above provides direct pipeline-level validation of this connection: the score used in the implemented progressive stage is not incidental, but **predictive of which generated samples actually improve target-domain generalization**.
> > >
> > > We hope this fully addresses your concern, and we would be grateful if you would consider reflecting this in your final assessment. We will properly include all the rebuttal concerns in the revised version, following your valuable suggestions. Please let us know if you have any further questions.

---

### Official Review · Reviewer_iyFp · 2026-03-12

**Soundness:** 3
**Presentation:** 3
**Significance:** 2
**Originality:** 2
**Overall Recommendation:** 4
**Confidence:** 5

**Summary:**

This paper studies single-domain generalization (SDG) for time series classification, where a model must generalize to unseen domains despite being trained on data from only a single source environment. The authors propose CURE (Context-driven Diffusion with Progressive Expansion), a framework that generates synthetic time series data using a conditional diffusion model guided by disentangled semantic-aware and semantic-free contexts.

The method first decomposes time-series representations into invariant semantic features (capturing class information) and semantic-free features (capturing domain-specific variation). These representations are then used as conditioning signals for a diffusion model that generates new samples with preserved class semantics but diversified domain styles. The framework further introduces a progressive expansion strategy, which maintains a memory bank of domain prompts and filters generated samples near decision boundaries to stabilize training.

Experiments are conducted on four human activity recognition datasets (EMG, DSADS, PAMAP2, and USC-HAD). Results show consistent improvements over several baseline domain generalization methods. Ablation studies examine the contribution of the disentanglement losses, adversarial objectives, and filtering strategy.

**Compliance With Llm Reviewing Policy:**

Affirmed.

**Final Justification:**

I appreciate the authors' detailed rebuttal and follow-up responses. As my concerns have been addressed, I support the acceptance of this manuscript.

**Key Questions For Authors:**

1. The theoretical analysis focuses primarily on the contrastive objectives. Could the authors clarify how the effectiveness of the diffusion-based augmentation mechanism improves domain generalization?

2. The framework relies on a conditional diffusion model to generate synthetic time-series data. What is the computational overhead of training this model compared to simpler augmentation methods?

3. The similarity operator used in several equations (e.g., feature comparison and nearest-neighbor retrieval) is denoted as ⋆. Is this implemented as cosine similarity, dot product, or another metric? Clarifying this would improve reproducibility.

**Limitations:**

No.

**Strengths And Weaknesses:**

## Strengths

1. The paper addresses an important problem: single-domain generalization for time-series classification, which is highly relevant in settings where collecting multi-domain labeled data is difficult.

2. The proposed framework combines representation disentanglement, contrastive learning, and conditional diffusion-based generation into a coherent architecture to synthesize diverse domain variations while preserving class semantics.

3. The progressive expansion mechanism, which maintains a memory bank of domain prompts and filters boundary samples, is an interesting strategy to gradually expand domain diversity while stabilizing training.

4. The paper is generally well-structured and readable, with a modular presentation separating representation disentanglement, diffusion-based generation, and progressive expansion. Figures illustrating the pipeline and t-SNE visualizations help convey the intuition behind the method.

5. Experimental results show consistent improvements over multiple baseline domain generalization methods, and the ablation studies provide evidence that key components of the framework contribute to performance gains.

6. The idea of using disentangled contexts as conditioning signals for diffusion-based data generation represents a creative combination of existing techniques in representation learning and generative modeling.

## Weaknesses

1. The theoretical analysis is relatively weak and does not directly justify the full learning procedure. The theorems mainly analyze properties of the contrastive objectives rather than the diffusion-based generation process or the progressive expansion mechanism. The connection between theory and empirical performance remains unclear.

2. Several implementation details are insufficiently specified, such as:
   - the exact similarity function used in retrieval and contrastive losses
   - diffusion model architecture and training settings
   - computational cost and training overhead of diffusion-based generation

3.  The methodological novelty is somewhat incremental, as the framework mainly combines existing techniques from representation learning and diffusion generative modeling.

---

> ### Author Rebuttal · Authors · 2026-03-31
>
> We are truly grateful for the time you have taken to review our paper, your insightful comments and support. Your positive feedback is incredibly encouraging for us! In the following response, we would like to address your major concern and provide additional clarification.
>
> > **Q1&Q4.** Theorems analyze.
>
> A1. Thanks for your comment! To further clarify the role of the diffusion-based generation component in our framework, we now include an additional theorem showing that, for any classifier $h\in\mathcal H$,
> $$R(h)\le \hat{R}_M(h)+L_1\Delta_s+L_2\varepsilon.$$
> This result directly connects the diffusion-based augmentation mechanism to target-domain generalization: $\Delta_s$ measures how well the expanded memory bank approximates an unseen domain factor, while $\varepsilon$ characterizes the fidelity of the diffusion output. Therefore, strong performance on the augmented distribution $\hat{X}_M$ transfers to the target domain whenever the bank coverage improves and the generation error remains small. We will clarify this interpretation in the revised version.
> ## Additional theorem
> We consider a latent factorization of time series data as
> $$X=F(c,s,\Theta),\qquad Y=\Psi(c),$$
> where $c, s,$ and $\Theta$ are semantic-aware, domain-specific, and latent factors. In practice, it is impossible to observe $\Theta$, thus it is reasonable to assume
> $$X \sim p( \cdot | c,s ) .$$
> For an unseen target domain $s$, define the target risk
> $$R(h)=\mathbb E\_{X}\big[\ell(h(X),Y)\big],$$
> where $h \in \mathcal{H}$ is the class discriminator and $l$ is the classification loss in (7). Recall the current prompt memory bank after $M$ steps is
> $$\mathcal{S}\_M=\{z\_1,\dots,z\_M\}.$$
> For any $s$, define its nearest prompt in the memory bank as
> $$\hat{s}\_M(s)=\arg\min\_{z\in\mathcal S\_M}\|s-z\|\_2,$$
> use $\Delta\_s = \mathbb{E}\big(\hat{s}\_M(s)\big)$ to denote its expectation. Let the augmented risk based on memory bank be
> $$\hat{R}\_M(h)=\mathbb E\_{\hat{X}\_M}\big[\ell(h(\hat{X}\_M),Y)\big].$$
> We introduce two assumptions before proposing the theorem
>
> **Assumption (A.1)** For any $h\in\mathcal H$, let $H\_1(c,s,\Theta) = l\big( h(F(c,s,\Theta)), \Psi(c) \big)$ and $H\_2(x,c) = l\big( h(x),\Psi(c) \big)$. $H\_1$ is $L\_1$-Lipschitz w.r.t. $s$, which means for any domain-specific feature $s,s'$,
> $$\big| H\_1(c,s,\Theta) -  H\_1(c,s',\Theta)\big| \le L\_1\|s-s'\|\_2,$$
> and $H\_2$ is $L\_2$-Lipschitz w.r.t. $x$,
> $$\big| H\_2(x,c) -  H\_2(x',c)\big| \le L\_2\|x-x'\|\_2.$$
> **Assumption (A.2)** For each pair $(c,s)$, we recall
> $$\hat{X}\_M \sim p\big(\cdot\mid c, \hat{s}\_{M}(s)\big)$$
> as the bank-induced diffusion output and let
> $$\widetilde{X}\_M = F(c, \hat{s}\_M,\Theta)$$
> be the ideal prompt-conditioned output. There exists constant $\varepsilon > 0$ such that
> $$\mathbb{E}\_{\hat{X}\_M} \big[ \| \hat{X}\_M - \widetilde{X}\_M  \|\_2\big] \leq \varepsilon.$$
>
> Condition (A.1) ensures stability w.r.t. latent factors. (A.2) is natural as bank expansion improves data representation and compresses the gap between diffusion and ideal outputs over iterations.
>
> **Theorem: Domain generalization gain from diffusion-based generation**
>
> Assume conditions (A.1) and (A.2). Then, for any $h\in\mathcal H$,
> $$R(h)\le \hat{R}\_{M}(h) + L\_1 \Delta\_s + L\_2 \varepsilon.$$
>
> We will add this and the proof in the revised version.
>
> > **Q2&Q5&Q6.** Implementation details:
> >   - (Q6)similarity function
> >   - diffusion model settings
> >   - (Q5)computational cost of diffusion-based generation
>
> A2. Thanks for your comment. We provide the missing implementation details below:
> - **Similarity function.** We use cosine similarity in both nearest-neighbor retrieval and the contrastive losses.
> - **Diffusion model architecture and training settings.** The diffusion generator is a conditional 1D U-Net with base width 64 and channel multipliers (1, 2, 4, 8). We train it with a DDPM objective, 100 diffusion steps, 25 sampling steps, AdamW optimizer, learning rate 2e-4, and batch size 32.
> - **Computational cost and training overhead.** We would like to clarify that the diffusion generator in our framework is trained only once and then reused for subsequent progressive generation. To reflect the practical cost in the progressive generation setting, we compare diffusion-based method against a GAN-based alternative. As shown below, the diffusion-based design achieves **comparable efficiency** in practice while also obtaining better performance.
>
> |Method|Runtime/iteration|EMG|
> |-|-:|-:|
> |GAN|44.05s|72.68|
> |Diffusion|20.79s|77.69|
>
> We will clarify these details in the revision.
>
> > **Q3.** Incremental novelty.
>
> A3. Thanks for your comment! We position the contribution of CURE as a coordinated design for single-source TSC, rather than a new standalone primitive. A more detailed clarification is provided in response to **Q1 of Reviewer xSiq**.
>
> Thanks again for appreciating our work and for your constructive suggestions. Please let us know if you have further questions.

---

> > ### Author Rebuttal · Reviewer_iyFp · 2026-04-03
> >
> > Thank you for the rebuttal. While the response provides some helpful clarification, my main concern remains only partially addressed. In particular, the added theorem still appears to be a high-level risk bound built on additional assumptions, rather than a result that directly analyzes the actual progressive expansion mechanism, memory bank updating, boundary filtering, and the full optimization pipeline. As a result, the core gap I raised remains. Overall, my assessment remains unchanged, and I support acceptance of the paper.

---

> > > ### Author Response · Authors · 2026-04-06
> > >
> > > Thank you for your thoughtful follow-up question and for engaging closely with our rebuttal.
> > >
> > > The theorem is built on the two quantities that CURE explicitly manipulates during progressive expansion: **domain coverage** and **generated-sample quality**. **Memory-bank updating** changes the set of domain-style anchors used in subsequent generation, thereby changing coverage of the target style space. The **progressive score** ranks generated samples according to their relative position to the assigned domain center and competing domain centers, and retraining is performed on the selected subset rather than on indiscriminate synthetic data. Therefore, **the theorem is aligned with the actual mechanism** at the level of the quantities the mechanism directly controls.
> > >
> > > We further tested this mechanism directly with a controlled sample-selection experiment on EMG (where "All generated samples" uses the full synthetic set, "High-score subset only" retains only the samples that pass the progressive boundary_ratio filter, and "Low-score subset only" consists exclusively of the excluded ambiguous samples):
> > >
> > >   | Supplementary setting | Best Mean Acc |
> > >   |---|---:|
> > >   | All generated samples | 64.38 |
> > >   | Low-score subset only | 60.06 |
> > >   | High-score subset only | 77.69 |
> > >
> > > The **higher-score subset is substantially more effective** than both the lower-score subset and using all generated samples together. This is direct evidence that the progressive mechanism used in the implementation is meaningful and **causally connected to domain generalization performance**.
> > >
> > > We hope this fully addresses your concern, and thank you again for your valuable suggestions. We will properly include all the rebuttal concerns in the revised version, following your valuable suggestions. Please let us know if you have any further questions!

---

### Official Review · Reviewer_PrLo · 2026-03-13

**Soundness:** 3
**Presentation:** 2
**Significance:** 2
**Originality:** 2
**Overall Recommendation:** 4
**Confidence:** 2

**Summary:**

This paper studies single-domain generalization for time series classification. To address two main issues in single-domain generalization—namely, the lack of explicit semantic guidance during the augmentation process and the diversity collapse caused by low-quality augmented data—the authors propose Context-driven Diffusion with Progressive Expansion (CURE). CURE uses disentangled prompts as conditions for a diffusion model, which adversarially balances the trade-off between domain diversity and category features. By guiding the model to generate data from diverse distributions while preserving semantic consistency, the proposed approach achieves strong performance on most datasets.

**Compliance With Llm Reviewing Policy:**

Affirmed.

**Final Justification:**

Since the author has diligently completed the rebuttal to address my concerns, I will raise the score.

**Key Questions For Authors:**

1. In the Introduction, the objective of time series classification is described as “predicting future values of a time series.” However, this seems closer to the goal of time series forecasting. Could the authors clarify this point?
2. Do similar performance trends hold on datasets beyond activity recognition (e.g., machine fault detection datasets)?
3. In time-series research, each subject is often treated as a separate domain. However, in this work, the EMG dataset appears to divide 36 subjects into 4 domains. In this case, can the setting still be considered a true single-domain generalization problem?
4. The ablation study is conducted only on EMG and DSADS. Do similar trends hold for the other datasets (PAMAP2 and USC-HAD)?

**Limitations:**

yes

**Strengths And Weaknesses:**

Strengths:
1. The paper clearly defines the problem and systematically explains the motivations behind the proposed approach.
2. The authors release their code to support reproducibility. However, although the code contains many comments that help understand the structure, most of the comments are not written in English. In addition, the repository does not include a README.md, which makes it difficult to understand the overall structure of the codebase.
3. The methodology is explained in detail, and a thorough ablation study is conducted to evaluate the contribution of each component.

Weaknesses:
1. Most of the experimental datasets belong to the activity recognition domain.
2. The overview in Figure 1 is somewhat abstract, making it difficult to understand the entire architecture solely from the figure.
3. Although the paper claims to address generalization in a single-domain setting, the experiments group multiple subjects into a single domain. The specific grouping of subjects may influence the reported performance improvements.

---

> ### Author Rebuttal · Authors · 2026-03-31
>
> We are truly grateful for the time you have taken to review our paper and your insightful review. Here we address your comments in the following.
>
> > **Q1&Q5.** Most of the experimental datasets belong to the activity recognition domain. Do similar performance trends hold on datasets beyond activity recognition?
>
> A1. Thanks for your comment! Since most existing time-series domain generalization benchmarks are based on human activity recognition (HAR), we further adapted our method to an industrial anomaly detection task to assess its applicability beyond HAR scenarios. Specifically, we conducted experiments on MIMII-DG [1], a domain generalization benchmark derived from machine sound anomaly detection. Following [1], we used the autoencoder as the anomaly detection baseline, and then incorporated CURE-generated data into its training process. The AUC results demonstrate that CURE remains effective in this setting.
>
> |Method|MIMII-fan|MIMII-slider|
> |-|-:|-:|
> |Autoencoder|0.423|0.466|
> |Autoencoder + CURE-generated data|0.508|0.523|
>
> > **Q2.** The overview in Figure 1 is somewhat abstract, making it difficult to understand the entire architecture solely from the figure.
>
> A2. Thanks for your comment! The workflow of Figure 1 is as follows: in **Representation Disentanglement for Source Data Exploration**, each source sample is disentangled into a semantic-aware feature $c_i^c$ and a semantic-free feature $s_i^s$. In **Category Feature Retrieval and Conditional Generation**, retrieved category features together with category and domain prompts are used to condition the diffusion model for data generation. In **Domain-Prompt Memory Updating and Classifier Retraining**, the domain-prompt memory bank is progressively updated, and boundary samples are filtered before retraining the classifier. We will further revise Figure 1 to present this data flow more explicitly.
>
> > **Q3&Q6.** The EMG dataset appears to divide 36 subjects into 4 domains. In this case, can the setting still be considered a true single-domain generalization problem?
>
> A3. Thanks for your comment! We follow the standard split protocol in prior time-series DG benchmarks [2,3] to ensure fair comparison with existing methods. To further address your concern, we additionally conducted an extra experiment under a stricter subject-level split setting, where each individual subject is treated as one domain. Specifically, we selected 4 subjects and regarded them as 4 separate domains. This result still shows an advantage under the stricter subject-level split setting.
>
> |Method|T1|T2|T3|Avg|
> |-|-:|-:|-:|-:|
> |DI2SDiff|71.39|70.43|67.05|69.62|
> |SEED|77.02|71.68|68.90|72.53|
> |CURE|80.68|70.61|70.10|73.80|
>
> > **Q4.** In the Introduction, the objective of time series classification is described as "predicting future values of a time series."
>
> A4. Thank you for pointing this out. The phrase “predicting future values of a time series” is an imprecise description in the Introduction, as it corresponds to time-series forecasting rather than time-series classification. In our work, the task is to assign a class label to an observed time-series segment; forecasting, by contrast, aims to predict future signal values. We will revise the Introduction to correct this wording and make the distinction between these two tasks explicit.
>
> > **Q7.** The ablation study is conducted only on EMG and DSADS.
>
> A7. Thanks for your comment! The complete ablation table is in Appendix Table 5. Across PAMAP2 / USC-HAD, removing $L_{cat}$ reduces average accuracy by 13.79 / 32.40 points, removing $L_{dom}$ reduces it by 13.73 / 10.56, and removing sample filtering reduces it by 12.38 / 10.87. This confirms the same overall trend beyond EMG and DSADS.
> |Variants|EMG|DSADS|PAMAP2|USC-HAD|
> |-|-|-|-|-|
> |CURE w/o $\mathcal{L}_{ind}$|73.53|70.32|72.61|65.23|
> |CURE w/o $\mathcal{L}_{cat}$|62.99|68.91|62.87|37.26|
> |CURE w/o $\mathcal{L}_{dom}$|66.63|68.84|62.93|59.10|
> |CURE w/o Adversarial Tuning|61.83|69.64|72.68|63.17|
> |CURE w/o Sample Filtering|70.97|72.95|64.28|58.79|
> |CURE|77.69|74.23|76.66|69.66
>
> In light of these responses, we hope we have addressed your concerns, and hope you will consider raising your score. If there are any additional notable points of concern that we have not yet addressed, please do not hesitate to share them, and we will promptly attend to those points.
>
> **References**
>
> [1] Dohi K, Nishida T, Purohit H, et al. MIMII DG: Sound dataset for malfunctioning industrial machine investigation and inspection for domain generalization task[J]. arXiv preprint arXiv:2205.13879, 2022.
>
> [2] Lu W, Wang J, Sun X, et al. Out-of-distribution representation learning for time series classification[J]. arXiv preprint arXiv:2209.07027, 2022.
>
> [3] Qian H, Pan S J, Miao C. Latent independent excitation for generalizable sensor-based cross-person activity recognition[C]//Proceedings of the AAAI conference on artificial intelligence. 2021, 35(13): 11921-11929.

---

> > ### Author Rebuttal · Reviewer_PrLo · 2026-04-04
> >
> > Thank you for the detailed rebuttal. The additional experiment beyond HAR is useful, but I still think the evidence is somewhat limited since it is based on anomaly detection rather than another time-series classification benchmark. So I will keep my original score.

---

> > > ### Author Response · Authors · 2026-04-05
> > >
> > > Thank you for your continued engagement and the thoughtful follow-up. MIMII-DG is a time-series anomaly detection benchmark, which consists of temporal audio signals for industrial machine condition monitoring. To further address your concern, we have conducted an additional experiment on the cwrBearing dataset [1], a **machinery fault time-series classification benchmark**.
> > >
> > > **Dataset Description.** The cwrBearing dataset consists of vibration signals collected from drive end and fan end bearings under normal conditions and single-point defect conditions, sampled at 12,000 or 48,000 samples per second. It contains 4 domains, 4 classes, sequences of length 512, and 1 channel, representing a **machinery fault classification** scenario with clear domain shifts across different operating conditions.
> > >
> > > | Method | 0→1 | 0→2 | 0→3 | Avg |
> > > |---|---:|---:|---:|---:|
> > > | DDLearn | 68.20 | 31.58 | 67.37 | 55.72 |
> > > | Diversify | 85.41 | 73.38 | 70.92 | 76.57 |
> > > | SEED | 87.78 | 79.02 | 73.43 | 80.08 |
> > > | **CURE** | **88.10** | **80.46** | **81.93** | **83.50** |
> > >
> > > CURE achieves the best average accuracy, outperforming the strongest baselines. This demonstrates that CURE's effectiveness extends beyond HAR to a genuine time-series **classification** task in an industrial machinery context.
> > >
> > >
> > > We hope this fully addresses your concern, and we would be grateful if you would consider reflecting this in your final assessment. We will properly include all the rebuttal concerns in the revised version, following your valuable suggestions. Please let us know if you have any further questions.
> > >
> > > **Reference.**
> > >
> > > [1] Zhang S, Zhang S, Wang B, et al. Deep learning algorithms for bearing fault diagnostics—A comprehensive review[J]. *IEEE Access*, 2020, 8: 29857–29881.

---

### Decision · Program_Chairs · 2026-04-30

**Decision:**

Accept (regular)

**Comment:**

This paper proposes CURE, a single-domain generalization framework for time series classification that disentangles representations into semantic-aware and semantic-free components to condition a diffusion model for diverse data synthesis, stabilized by a progressive memory bank with boundary-based filtering. All three reviewers converged on Weak Accept, praising the well-motivated design and consistent empirical gains, while sharing concerns about limited novelty (combining established techniques), weak theoretical grounding, HAR-centric evaluation, and missing implementation details.

The rebuttal added a formal risk bound connecting diffusion fidelity and memory bank coverage to generalization, provided controlled ablations isolating the progressive filtering as the key performance driver, extended experiments to machinery fault classification (cwrBearing) and anomaly detection (MIMII-DG), and clarified architectural and computational details. Residual concerns about novelty and theoretical depth remain but are not disqualifying. I recommend Weak Accept, contingent on incorporating the rebuttal experiments and improved documentation in the camera-ready version.